# Algorithmically Detected Rain-on-Snow Flood Events in Different Climate Datasets: A Case Study of the Susquehanna River Basin

Colin M. Zarzycki[1], Benjamin D. Ascher[1,*], Alan M. Rhoades[2], and Rachel R. McCrary[3]

[1]Department of Meteorology and Atmospheric Science, The Pennsylvania State University, University Park, PA, USA
[2]Climate and Ecosystem Sciences Division, Lawrence Berkeley National Laboratory, Berkeley, CA, USA
[3]NSF, National Center for Atmospheric Research, Research Applications Laboratory, Boulder, CO, USA
[*]now at Colorado State University

**Correspondence:** Colin M. Zarzycki (czarzycki@psu.edu)

**Abstract.** Rain-on-snow (RoS) events in regions of ephemeral snowpack – such as the northeastern United States – can be key drivers of cool-season flooding. We describe an automated algorithm for detecting basin-scale RoS events in gridded climate data by generating an area-averaged time-series and then searching for periods of concurrent precipitation, surface runoff, and snowmelt exceeding pre-defined thresholds. When evaluated using historical data over the Susquehanna River Basin (SRB), the technique credibly finds RoS events in published literature and flags events that are followed by anomalously high streamflow as measured by gage data along the river. When comparing four different datasets representing the same 21-year period, we find large differences in RoS event magnitude and frequency, primarily driven by differences in estimated surface runoff and snowmelt. Using dataset-specific thresholds improves agreement between datasets but does not account for all discrepancies. We show that factors such as meteorological forcing and coupling frequency as well as choice of land surface model play roles in how data products capture these compound extremes and suggest care is to be taken when climate datasets are used by stakeholders for operational decision-making.

## 1 Introduction

Rain-on-snow (RoS) events have been increasingly studied over the past few decades, yet such research is overwhelmingly focused on mountainous regions with well-defined seasonal snowpacks (Singh et al., 1997; McCabe et al., 2007; Wayand et al., 2015; Sterle et al., 2019; Musselman et al., 2018; Poschlod et al., 2020; Hatchett, 2021; Siirila-Woodburn et al., 2021b; Heggli et al., 2022; Yu et al., 2022; Brandt et al., 2022; Maina and Kumar, 2023; Haleakala et al., 2023). Correspondingly, there has been less focus in areas with more ephemeral snow cover, such as the northeastern United States (US), even though RoS events, a flavor of compound extreme (AghaKouchak et al., 2020), produce many cases of 'slow-rise' flooding – floods generally occurring more than six hours after the onset of the meteorological driver (Dougherty et al., 2021). Climatologically, RoS events in the northeastern US peak in late winter and spring (Ashley and Ashley, 2008; Villarini and Smith, 2010; Dougherty and Rasmussen, 2019; Wachowicz et al., 2020) and are key drivers of flooding in New England and the Atlantic side of Canada (Collins et al., 2014). Synoptic case study analyses of recent RoS events in the mid-Atlantic highlight inland-running extratropical cyclones that advect warm, moist air into the region as key dynamical drivers (Grote, 2021; Suriano et al., 2023).

Rapid snow ablation in ephemeral snow regions has been shown to have a strong correlation with increases in basin streamflow in the days following snowmelt (Suriano et al., 2020, 2023).

While climatological studies are critically important, event-level analysis has become increasingly valuable when communicating climate risks (Shepherd et al., 2018). One river basin in the northeastern US that has historically dealt with RoS events is the Susquehanna River Basin (SRB) – a basin home to more than four million people (Leathers et al., 2008) and one considered climatologically flood-prone due to the wide variety of weather phenomena that occur within the region (Perry, 2000). The most consequential non-tropical-cyclone flood in recent SRB history was a RoS flood that occurred in January 1996, resulting in ∼$1.5 billion in damages and 30 fatalities (Leathers et al., 1998). Significant events such as this are frequently used by stakeholders as a point of reference for real-time forecasts and long-term planning (George and Mudelsee, 2019). Other evidence, such as the Great Flood of 1936 (another RoS event) and the fact that sediment records indicate prehistoric periods of high flood activity are associated with negative phases of the North Atlantic Oscillation (NAO) (which drive positive snowpack anomalies in the northeastern US, (Hartley and Keables, 1998)) underscore the importance of these events to regional hydrology (Toomey et al., 2019).

While studying historical events is important for planning purposes, and counterfactual reforecasts using imposed warming approaches can provide clues as to how similar events may unfold in the future (e.g., Pettett and Zarzycki (2023)), it is difficult for such approaches to provide risk quantification from a frequency-of-occurrence perspective. For these climatological evaluations, it is desirable to be able to identify such events in both observational datasets and model simulations, including future climate projections. Unfortunately, assessing extreme, compound, and discrete events in climate data sets is a complex challenge, particularly because such datasets are not necessarily developed specifically for this purpose (Angélil et al., 2017; Parker, 2020) and there is commonly a lack of observational reference datasets to quantify their fidelity. Gridded datasets (spatiotemporally continuous data provided on a regular latitude-longitude mesh) of the historical record are frequently used to assess hydrometeorological extremes in locations with poor or non-existent station observations and climate analyses requiring multiple complex variables in addition to evaluating model sensitivity and performance. Further, future changes in RoS event frequency and character due to climate change are projected via the use of free-running models, which operate on numerical grids and don't have an *a priori* record to compare to, making the use of an automated heuristic a requirement.

Therefore, to extract information regarding compound hydrometeorological extremes, such as RoS events, and their corresponding statistics, algorithmic techniques that objectively analyze datasets without manual intervention are desirable. Here, we demonstrate a technique for generating a RoS event database at the basin scale for arbitrary gridded datasets and intercompare key decision-relevant differences (e.g., flood frequency) across four climate data products. We choose to focus on the SRB based on its proximity to major population centers, existing evidence for increasing flood hazards and exposure in the basin (Sharma et al., 2021), and the aforementioned 1996 extreme event serving as a benchmark for the mid-Atlantic US, although the technique described can be applied to any geographically defined basin with properly specified thresholds.

## 2 Methods

### 2.1 Datasets

We evaluate RoS events within three widely-used climate datasets and in one state-of-the-art Earth system model (ESM) nudged towards an atmospheric reanalysis. All four datasets seek to reproduce observed conditions, although each uses a distinct methodology to do so. First, we investigate the dataset described in Livneh et al. (2015) (hereafter, L15). L15 is a widely-used 1/16° hydrometeorological dataset covering most of North America. Meteorological data provided by L15 consists of daily precipitation, temperature (maximum and minimum), and wind speed at each location (Henn et al., 2018). This data is then temporally interpolated to obtain subdaily estimates (Bohn et al., 2013), which are then used to drive the Variable Infiltration Capacity (VIC, Liang et al. (1994)) land surface model (LSM) to produce hydrometeorological outputs. Next, we investigate the 1/8° North American Land Data Assimilation System (NLDAS-VIC4.0.5) described in Xia et al. (2012). NLDAS is driven by offline atmospheric forcing derived from the North American Regional Reanalysis, with adjustments made to some variables based on observations. NLDAS uses a combination of daily observations and radar data to produce hourly estimates of precipitation. While there are multiple NLDAS-2 LSMs that produce hydrologic output variables, we analyze only the NLDAS-VIC4.0.5 as it is the most methodologically consistent with L15. We also analyze a ∼0.5° global reanalysis (JRA-55, Kobayashi et al. (2015)), generated by running a prognostic ESM while continually assimilating observational data during integration. Global reanalyses serve as a bridge between in-situ, but spatio-temporally unstructured, observational data and free-running climate models (Parker, 2016). Finally, a ∼1° nudged version of the U.S. Department of Energy (DOE) Energy Exascale Earth System Model (E3SM, Golaz et al. (2022)) model is analyzed (Zhang et al., 2022). Assessing this dataset provides insight into the capability of ESMs used in climate assessments (e.g., the Coupled Model Intercomparison Project Phase 6 (CMIP6), Eyring et al. (2016)) to capture observed hydrometeorological extreme events. E3SM performs close to the CMIP6 ensemble mean across several measures of skill (Fasullo, 2020). To constrain the large-scale meteorology in E3SM to match observed conditions, the E3SM simulation is nudged using 6-hourly fields from the ERA5 reanalysis (Hersbach et al., 2020) generated using the technique contained in the Betacast toolkit, initially outlined in Zarzycki and Jablonowski (2015). This nudging acts as a crude assimilation technique and is only applied in the free atmosphere (above nominally 850 hPa, or 1 kilometer). The zonal and meridional winds are nudged to ERA5 analysis with a relaxation timescale $\tau$ = 3 hours and the temperature is nudged with $\tau$ = 24 hours. No nudging is performed on the surface pressure or moisture fields. The simulation reproduces observed patterns of 500 hPa geopotential height and sea level pressure while simulating grid-scale processes relatively untethered by the nudging reanalysis product (Sun et al., 2019). E3SM is run with a spectral element dynamical core on an unstructured cubed-sphere mesh ($n_e 30 n_p 4$) and simulation output is remapped to a 1°x1° rectilinear grid using higher-order methods (Hill et al., 2004). All model settings and tunings are left as the default contained in the commit used here (f9cbe57).

## 2.2 Defining basin-scale events

A schematic of the RoS detection algorithm described here is shown in Fig. 1. To identify RoS events in the datasets, three hydrometeorological variables are used; precipitation (PRECIP, includes both rainfall and snowfall), surface water runoff (ROF), and snow water equivalent (SWE). The 24-hour change in SWE from the previous day (dSWE) is calculated via a simple backward difference. All data is standardized to daily average values (00Z to 00Z) by temporally averaging any sub-daily (e.g., hourly) data at each grid cell. Grid cells that have at least 50% of their area enclosed by the shapefile boundary of a river basin (here, the SRB) are kept, while those exterior to the basin are set to missing values (Fig. 1a). The resulting SRB domains for each product can be seen in Fig. 2. A basin-wide time series is then constructed by spatially averaging fields for each day and smoothing the resulting time series using a moving average to reduce day-to-day noise. We choose a 5-day window based on mid-latitude synoptic timescales (Holton, 2004), although other window durations did not materially impact these findings (not shown).

This results in a one-dimensional time series with a single value for each calendar day that represents the basin-averaged conditions for each data product. RoS days are then defined by flagging periods of positive ROF and negative dSWE (snow loss) that both exceed specified thresholds ($t$). We test two methods of thresholding; one uses fixed thresholds across all four datasets (FIXED) and the other defines dataset-specific thresholds by those exceeding 95% of all daily values (RELATIVE). For FIXED, we require an average ROF of 1.4 mm/day ($t_{ROF}$) and dSWE of -1.4 mm/day ($t_{dSWE}$) averaged across the basin based on a manual sensitivity analysis. Thresholds in the RELATIVE configuration are calculated independently for each product by computing the 95% percentile value for each variable in the unsmoothed daily 1985-2005 time series (Fig. 1b). This results in an array containing a binary value for each day defining whether or not it is a 'RoS day.' Contiguous RoS days are considered to be part of a single 'RoS event' such that discrete events can last for different durations.

To enforce a criterion that precipitation occurs during at least some portion of the event, we require an average PRECIP of 2 mm/day ($t_{PRECIP}$), which is kept the same in both the FIXED and RELATIVE configurations. We note that our findings are actually largely insensitive to the magnitude of the PRECIP threshold (or even its inclusion), implying the vast majority of events with high dSWE and ROF are also associated with non-zero PRECIP. This supports the notion that similar synoptic meteorological patterns (Grote, 2021) and additional liquid input to the surface (i.e., runoff being a combination of snowmelt and water flux from the atmosphere) lead to the majority of RoS floods being associated with, at least, some precipitation versus precipitation alone being a primary driver of such events. It also concurs with Suriano et al. (2023), who found that, while RoS days contributed disproportionately to extreme snow ablation events, snowmelt also occurred in a myriad of non-precipitating patterns, including high-pressure overhead and under northwesterly/westerly large-scale flow. We argue this also corroborates the finding that actual heat transfer between the liquid rain and the surface of the snowpack is rather small and explains only a small fraction of the observed snowmelt (Moore and Owens, 1984).

A sample RoS event detection is shown in Fig. 1c. The smoothed basin-averaged daily time series of PRECIP, ROF, and dSWE are shown from top-to-bottom in dark green, blue, and red respectively (the thinner line represents the raw, unsmoothed time series). The various thresholds ($t_{PRECIP}$, $t_{ROF}$, and $t_{dSWE}$) are shown as dashed horizontal lines. The area where the metric

exceeds the relevant $t$ (i.e., days that the variable's RoS criterion is satisfied) is shaded for each time series. The vertical black lines denote the start and end of the event, defined by the first and last times when all three quantities exceed their defined threshold $t$. Gray shading represents contiguous days where all three criteria are satisfied, thus defining a RoS event. Here, an event from January 17th to January 25th, 1996 was added to the record.

125    For context, our definition here of RoS is somewhat arbitrary. Most studies enforce some fixed combination of precipitation and snowpack threshold. Ye et al. (2008) classified events as liquid precipitation falling onto at least 10 mm of existing snowpack in a gridbox. Musselman et al. (2018) define 'RoS days' where at least 10 mm of precipitation falls on at least 10 mm of snowpack in a given day. This technique was adopted by López-Moreno et al. (2021) and Maina and Kumar (2023). A similar strategy, but applying higher thresholds (20 mm precipitation, 250 mm snowpack), was used by Würzer et al. (2016). Likewise,

130    Hotovy et al. (2023) required 10 mm of existing SWE and greater than 0°C surface temperatures to enforce collocated daily precipitation (of at least 5 mm) to fall in liquid form. Instead of a SWE metric, others have applied snow cover fraction or some other snow/no-snow classification in addition to a precipitation threshold (Mazurkiewicz et al., 2008; Pradhanang et al., 2013; Cohen et al., 2015). Others include some measure of snowmelt, such as Freudiger et al. (2014) and Li et al. (2019) who required 3 mm of rainfall, 10 mm SWE, and no larger than a 4:1 ratio between rainfall and snowmelt in order to eliminate rain-only

135    events (Musselman et al. (2018) also includes the latter requirement). Suriano (2022) similarly required a 10 mm daily snow depth decrease that occurred in combination with above-freezing temperatures and non-zero precipitation. However, relative strategies have also been proposed, such as the 98% threshold used for covarying extremes in the compound event analysis of Poschlod et al. (2020). Other studies simply require the joint occurrence of precipitation and snowmelt in a given period (McCabe et al., 2007; Surfleet and Tullos, 2013; Collins et al., 2014; Guan et al., 2016; Jeong and Sushama, 2018). Wachowicz

140    et al. (2020) used the same gridded data and explored four different definitions derived from some of the above and found high-level agreement.

This brief literature review is not meant to be considered exhaustive, but rather to highlight that both RoS evaluation techniques in this manuscript fall within the scope of existing strategies. It also emphasizes that extreme RoS events in mountainous regions with more seasonal snowpack or those using gridpoint values versus basin-integrated metrics may require different

145    thresholds. With the caveat that much of the previously cited RoS work has focused on regions with less ephemeral snowpack (e.g., western U.S. mountains), the algorithm discussed here falls within the envelope of previously published results both from a heuristic perspective and also with respect to our defined thresholds. We also note that our decision to use surface runoff in our definition is unique and implicitly includes the permeability of soil in our calculation. For example, the 1996 SRB event was exacerbated by first frozen, and then saturated soils, allowing for significant lateral movement of water across the surface

150    (Yarnal et al., 1997). While the application of this threshold is intuitive, we acknowledge it does introduce the potential for land surface errors to be more apparent in RoS detection (e.g., biases in soil properties in conditions).

Lastly, to show all datasets adequately represent the same period, Pearson correlations of basin-wide statistics (Table 1) were statistically significant with a two-sided $t$-test between all permutations of daily time series. This confirms that all datasets fundamentally represent the same meteorology and land surface evolution as processed here and can therefore be directly

155    compared to one another.

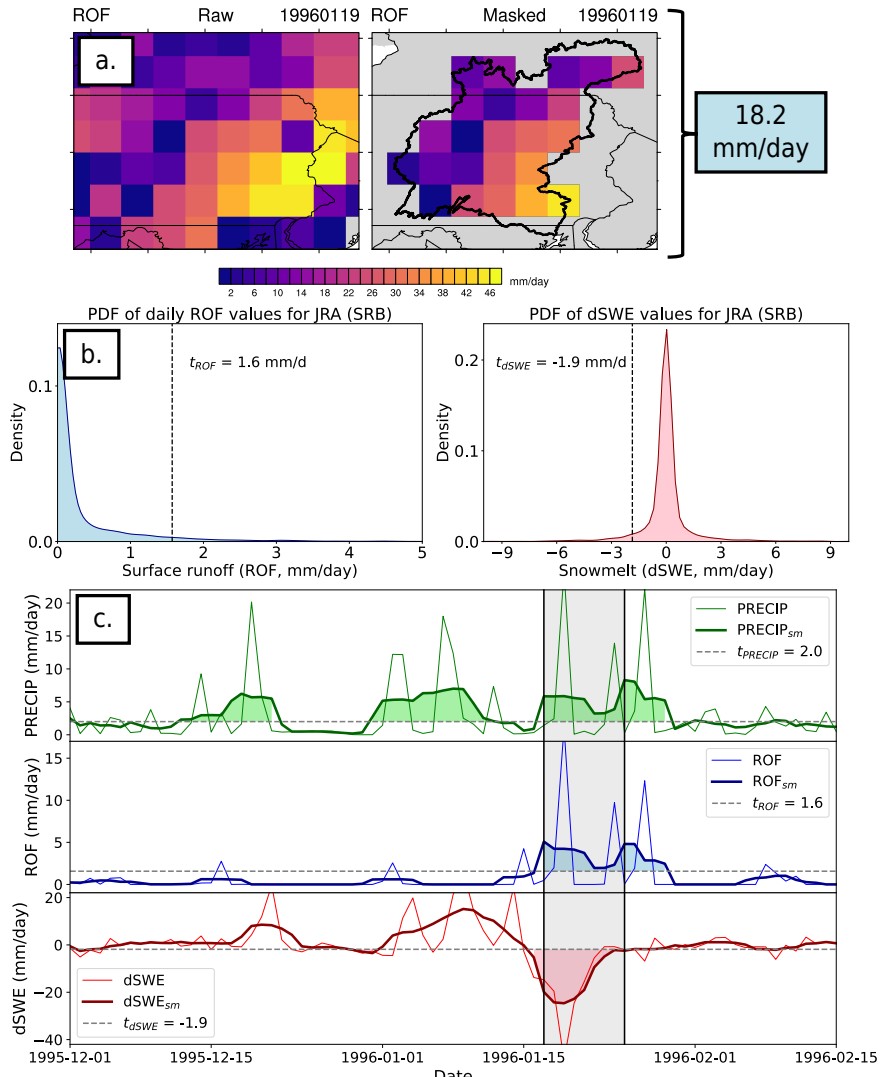

**Figure 1.** Schematic demonstrating how RoS events are defined in this work. (a.) Gridded daily-average data is first masked to only retain data within a defined shapefile (here the SRB) and then area-averaged to produce a single value on that day for the area. (b.) Exceedance thresholds $t$ can be computed from these daily values by using a specified percentile (e.g., 95%) of the distribution of the entire dataset. (c.) Finally, RoS events are defined as contiguous days where the basin-averaged time series of ROF, dSWE, and PRECIP all exceed their thresholds $t_{ROF}$, $t_{dSWE}$, and $t_{PRECIP}$, respectively. In the bottom plots, the darker lines represent the smoothed timeseries while the thinner lines denote the raw daily data. Shaded areas indicate periods when the given variable's time series is above its relevant threshold $t$ (denoted as a horizontal dashed line).

**Table 1.** Pearson correlations between data products for daily timeseries of precipitation (top), surface runoff (middle), and 24-hour change in snow water equivalent (bottom) over the study period. All values are correlated significantly at greater than the 99.9% confidence level using a two-sided $t$-test, indicating that all timeseries represent the same historical period and include relevant day-to-day variations over the SRB.

| | PRECIP | | | |
| --- | --- | --- | --- | --- |
| | JRA | L15 | NLDAS | E3SM |
| JRA | 1.00 | 0.76 | 0.89 | 0.75 |
| L15 | 0.76 | 1.00 | 0.81 | 0.93 |
| NLDAS | 0.89 | 0.81 | 1.00 | 0.77 |
| E3SM | 0.75 | 0.93 | 0.77 | 1.00 |
| | ROF | | | |
| | JRA | L15 | NLDAS | E3SM |
| JRA | 1.00 | 0.89 | 0.88 | 0.83 |
| L15 | 0.89 | 1.00 | 0.85 | 0.74 |
| NLDAS | 0.88 | 0.85 | 1.00 | 0.82 |
| E3SM | 0.83 | 0.74 | 0.82 | 1.00 |
| | dSWE | | | |
| | JRA | L15 | NLDAS | E3SM |
| JRA | 1.00 | 0.34 | 0.46 | 0.47 |
| L15 | 0.34 | 1.00 | 0.79 | 0.66 |
| NLDAS | 0.46 | 0.79 | 1.00 | 0.71 |
| E3SM | 0.47 | 0.66 | 0.71 | 1.00 |

## 3 Results

### 3.1 Climatology

We first investigate the mean climatology of relevant quantities over the SRB to provide some context into each data product's baseline. The average cool-season (November-April, inclusive) distributions of PRECIP, ROF, SWE, and dSWE are shown in Fig. 2. The higher resolution of L15 and NLDAS is evident from the added structure in the mean fields of the first two columns. Mean PRECIP (Fig. 2a-d) is higher in JRA and E3SM when compared to the L15 and NLDAS. This may be due to factors such as atmospheric model biases or the inclusion of rain gauge data in L15 and NLDAS, although it has been shown that significant spread exists in historical gridded climate data products, even for more commonly observed variables such as precipitation (Gutmann et al., 2012; Livneh et al., 2014; Henn et al., 2018). ROF (Fig. 2e-h) and SWE (Fig. 2i-l) climatologies differ between the data products. Notably, L15 produces mean ROF values that are less than half of the climatologies of each of the other three datasets. It is well-known that simulated ROF from different hydrologic models can be extremely variable,

with regional differences between products reaching an order of magnitude (Gudmundsson et al., 2012; Sood and Smakhtin, 2015; Beck et al., 2017). NLDAS produces less SWE climatologically, <20% of the SWE produced by L15 or E3SM, and even less than the coarser JRA product. Previous work has also shown that SWE estimates can vary greatly across datasets (Lundquist et al., 2015; Rhoades et al., 2018), particularly over the ephemeral snow area of the northeastern U.S. (McCrary et al., 2017, 2022). dSWE climatology is shown in Fig. 2m-p. When calculating the mean, all accumulation (or zero change) days are ignored to isolate only days where snow loss occurred. Here, NLDAS also exhibits the lowest magnitude of dSWE, although we speculate this is at least partly due to the shallower mean snowpack. The largest climatological dSWE magnitude is found in JRA, even though the SWE does not contain the largest depths, implying the snowpack is more variable in JRA and may be prone to more rapid snow loss (from a dSWE per unit time perspective).

We emphasize that, from a physical standpoint, differences in snowfall and snowmelt timing (Rauscher et al., 2008; McCabe and Clark, 2005), snow properties (Brown et al., 2006), temperature and permeability of the soil (Niu and Yang, 2006), precipitation type partitioning (Knowles et al., 2006), and evapotranspiration (Zheng et al., 2019) all can lead to differences in how these quantities are simulated. We speculate that these mechanisms are playing important roles in the differing mean climatologies but performing a fully-detailed water budget analysis for each of the land surface models leveraged by these datasets is beyond the scope of this analysis.

## 3.2 Flagged event statistics

Turning to the algorithmically flagged RoS events, Table 2 shows summary statistics for all four data products. Focusing on the FIXED thresholds, the total number of discrete events that occurred in the SRB over the 21-year study period varies by an order of magnitude, from 6 events in L15 to 58 in E3SM. Interestingly, large differences don't necessarily appear when considering the mean event-averaged dSWE, ROF, or PRECIP (last three columns). In fact, when a RoS event occurs, L15 has the largest $PRECIP_a$ and largest $dSWE_a$, although the smallest $ROF_a$.

When using the RELATIVE thresholds, the event frequencies agree better, with only a factor of two difference between L15/NLDAS and JRA/E3SM. The number of events in L15 increases because the ROF threshold $t_{ROF}$ (95th percentile of daily values) is reduced from 1.4 to 0.6 mm/day. Similarly, the dSWE threshold $t_{dSWE}$ magnitude is reduced from -1.4 mm/day to -0.8 mm/day in NLDAS. Conversely, fewer events were classified in E3SM, due to increases in both the required ROF and dSWE thresholds applied to the daily climatology. However, even when accounting for the baseline climatological differences of the data products by thresholding on percentiles, rather than absolute magnitudes, differences still are evident in all metrics.

We also perform a sensitivity analysis intended to include the requirement in Freudiger et al. (2014) and Musselman et al. (2018) that the sum of rainfall and snowmelt contains at least 20% snowmelt. This is added as an additional threshold $t_{fSWE}$ = 0.2 where fSWE (fraction of dSWE contribution) is computed as dSWE divided by the sum of dSWE and PRECIP smoothed time series (note, the sign of dSWE is inverted to be a positive contributor to liquid water on the surface). We refer to this simulation as RELATIVE_F14 since it preserves the same thresholds in RELATIVE with this one additional exclusionary check from Freudiger et al. (2014) in order to remove high rainfall (but low snow loss) events. The number of events is the same for all datasets except NLDAS, which loses 7 events over the study period when enforcing $t_{fSWE}$ = 0.2. This can be

**Table 2.** Statistics of RoS events over the SRB using FIXED thresholds (top) and RELATIVE thresholds (bottom). $t_{\mathrm{PRECIP}}$, $t_{\mathrm{ROF}}$, and $t_{\mathrm{dSWE}}$ represent the thresholds used for precipitation, surface runoff, and snow water equivalent loss, respectively (mm/day). Events represent the number of RoS events flagged over the 1980-2005 period. Duration is the average number of consecutive days an RoS event lasts. $\mathrm{PRECIP}_a$, $\mathrm{ROF}_a$, and $\mathrm{dSWE}_a$ represent the amount of precipitation rate, amount of runoff rate, and average snow loss (mm/day) per event by calculating the mean daily value for each individual event and then averaging those.

| | $t_{\mathrm{PRECIP}}$ | $t_{\mathrm{ROF}}$ | $t_{\mathrm{dSWE}}$ | $t_{\mathrm{fSWE}}$ | Events | Duration | $\mathrm{PRECIP}_a$ | $\mathrm{ROF}_a$ | $\mathrm{dSWE}_a$ |
| | $mm\ day^{-1}$ | $mm\ day^{-1}$ | $mm\ day^{-1}$ | $\%$ | $\#$ | $days$ | $mm\ day^{-1}$ | $mm\ day^{-1}$ | $mm\ day^{-1}$ |
|---|---|---|---|---|---|---|---|---|---|
| FIXED | | | | | | | | | |
| L15 | 2.0 | 1.4 | -1.4 | - | 6 | 3.2 | 9.5 | 1.5 | -5.3 |
| NLDAS | 2.0 | 1.4 | -1.4 | - | 16 | 2.9 | 6.3 | 2.1 | -5.2 |
| JRA | 2.0 | 1.4 | -1.4 | - | 48 | 3.5 | 5.3 | 2.6 | -4.4 |
| E3SM | 2.0 | 1.4 | -1.4 | - | 58 | 4.1 | 6.5 | 2.4 | -5.2 |
| RELATIVE | | | | | | | | | |
| L15 | 2.0 | 0.6 | -1.5 | - | 20 | 5.2 | 6.2 | 1.1 | -5.3 |
| NLDAS | 2.0 | 1.4 | -0.8 | - | 20 | 2.8 | 7.2 | 2.1 | -4.2 |
| JRA | 2.0 | 1.6 | -1.9 | - | 40 | 3.1 | 5.1 | 2.5 | -4.1 |
| E3SM | 2.0 | 1.8 | -2.2 | - | 41 | 4.1 | 6.9 | 2.9 | -8.0 |
| RELATIVE_F14 | | | | | | | | | |
| L15 | 2.0 | 0.6 | -1.5 | 20 | 20 | 4.9 | 6.0 | 1.0 | -5.3 |
| NLDAS | 2.0 | 1.4 | -0.8 | 20 | 13 | 3.0 | 4.4 | 1.9 | -5.1 |
| JRA | 2.0 | 1.6 | -1.9 | 20 | 40 | 3.0 | 5.3 | 2.4 | -4.2 |
| E3SM | 2.0 | 1.8 | -2.2 | 20 | 41 | 4.0 | 6.8 | 2.8 | -7.8 |

explained by the results in Fig. 2. NLDAS produces a 'wetter' precipitation climatology (Fig. 2f) but less climatological SWE (Fig. 2j) and, correspondingly, less dSWE (Fig. 2n). Therefore, enforcing a check that removes high PRECIP, low dSWE events would reduce events detected in NLDAS most strongly relative to other datasets. While the other three products have the same number of events with or without the inclusion of $t_{\mathrm{fSWE}} = 0.2$, the mean duration is slightly shortened, and mean event ROF is somewhat reduced using RELATIVE_F14, implying that a handful of high PRECIP, low dSWE (and high ROF) days at event onset or termination are lost. However, this reduction is small and therefore provides confidence that just including threshold checks for dSWE, ROF, and PRECIP is satisfactory for detecting RoS events over the SRB without a more formal snow loss cutoff. In the remainder of this paper, we omit the use of $t_{\mathrm{fSWE}}$ for simplicity. However, we want to emphasize that the simulation of land surface processes in different datasets can play a key role in precipitation/snowmelt partitioning, motivating further process-oriented evaluation of their joint occurrence in gridded climate data in the future. A percentile-based threshold-only algorithm (such as RELATIVE) may particularly struggle in regions of climatologically low SWE and high PRECIP (which experiences primarily rain-induced flooding) or high SWE and low PRECIP (primarily melt-induced flooding).

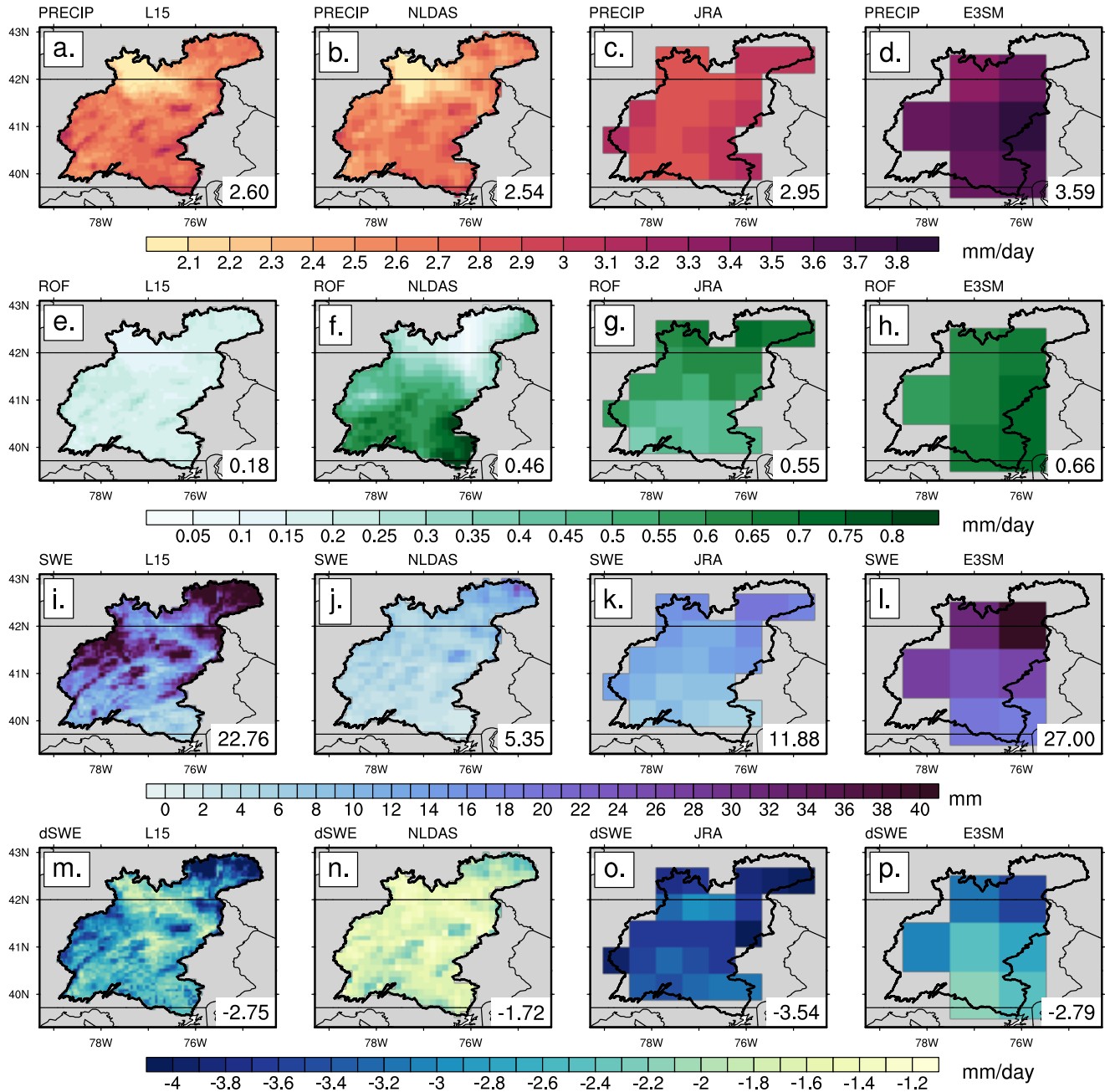

**Figure 2.** November to April (inclusive) mean climatologies of L15, NLDAS, JRA, and E3SM (left to right). From top to bottom are precipitation (PRECIP; mm/day), surface runoff (ROF; mm/day), snow water equivalent (SWE; mm), and daily change in snow water equivalent (dSWE; mm/day). dSWE only includes days with snow loss at a particular grid cell. Data is only included if >50% of a grid cell lies within the SRB bounds (black outline).

The differences across gridded climate data products are also shown in Fig. 3 (FIXED) and in Fig. 4 (RELATIVE). The sign of dSWE is flipped here for ease of visualization with the other metrics (i.e., positive dSWE in both figures denotes snow ablation). We focus our discussion on the relative threshold results in Fig. 4 for brevity, although note that the distributions of event statistics (Fig. 3b-d and Fig. 4b-d) are qualitatively similar between the two.

Figure 4a shows the total number of RoS events flagged over the climatological period (fourth column of Table 2), while Fig. 4b-d shows the frequency distribution for basin-averaged maximum rates of PRECIP, ROF, and dSWE at the event level. These are calculated by selecting the maximum daily value from the array of actual (i.e., unsmoothed) daily values during each RoS event.

The maximum PRECIP associated with flagged RoS events is similar between the four datasets, with E3SM tending to have slightly more extreme PRECIP occurring over the SRB (in agreement with Fig. 2). Much larger differences are seen in ROF and dSWE. In ROF, both the E3SM and JRA datasets produce larger event magnitudes of ROF, and subsequently have longer tails in their probability distributions, compared to the other two datasets. L15 produces RoS events with the least ROF (averaging approximately one-third of those found in either E3SM or JRA), with NLDAS in between the other three. The probability distribution functions of dSWE highlight an even more complex picture, with both NLDAS and L15 having narrower distributions with smaller magnitudes than E3SM. JRA and E3SM have broader distributions (i.e., longer tails) but the JRA distribution is more skewed, with frequent low dSWE events whereas E3SM has a more uniform distribution of dSWE rates.

Summarizing, using either a relative or fixed threshold to identify RoS events, the fully-coupled ESMs (E3SM and JRA) produce more events than L15 and NLDAS. While daily PRECIP in each product differs somewhat, it should be noted that these differences are relatively small. Rather, the majority of the differences in RoS events flagged arise from lower magnitudes of ROF and dSWE in L15 and dSWE in NLDAS. It is worth noting that L15 and NLDAS produce fewer RoS events regardless of which thresholding technique is used, so not only are the distributions of relevant daily variables shifted relative to the other models, but their skewness is impacted as well (Figs. 3 and 4).

### 3.3 Single-year evaluation

To further explore RoS events at a more granular level we focus on results associated with the 1996 water year (WY96, October 1995 to September 1996). We chose this WY since this includes the January 1996 SRB flood that is often used for disaster planning purposes in the region (U.S. Army Corps of Engineers, 2001). The same visualizations for other WYs are included in the data download associated with this manuscript.

Figure 5 shows the SRB daily WY96 time series of basin-integrated PRECIP, ROF, and dSWE. Negative (positive) dSWE denotes snowmelt (accumulation). Vertical shading denotes RoS events that were flagged by the algorithm (using the RELATIVE thresholds). At the top of each panel is a stripe with four different shadings, representing days where streamflow exceeded the 90th, 95th, 99th, and 99.9th relative percentiles over the 1985-2005 period at the US Geological Survey (USGS) gage at Harrisburg, Pennsylvania (USGS #01570500), with deepest blues and black representing highly anomalous flow conditions.

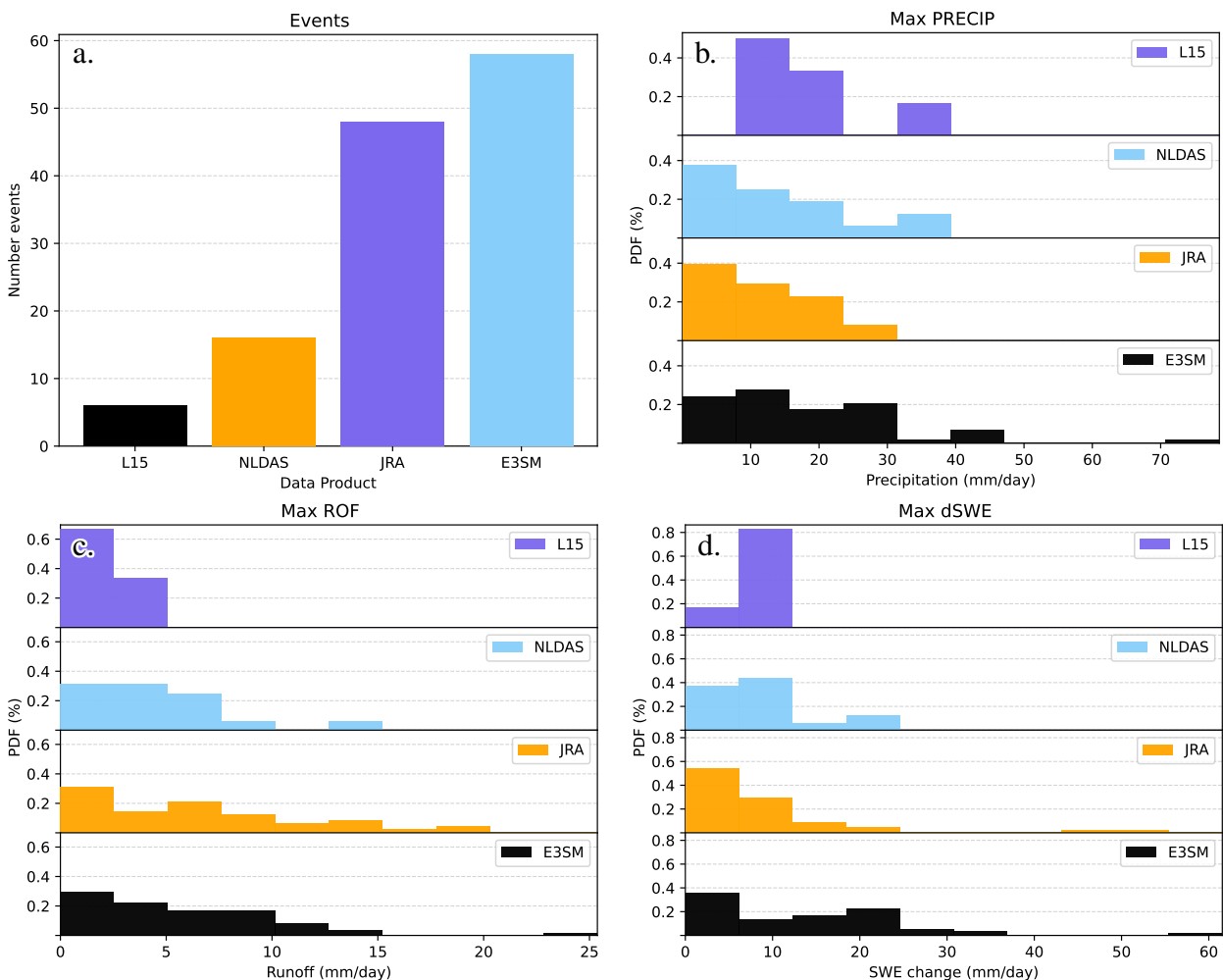

**Figure 3.** Histogram statistics for RoS events within the FIXED detection framework. The number of RoS events for each product from 1985-2005 is shown in the top left. The other three panels show frequency distributions of the daily rate (mm/day) of maximum precipitation (PRECIP), maximum runoff (ROF), and maximum change in SWE (dSWE, snowmelt positive).

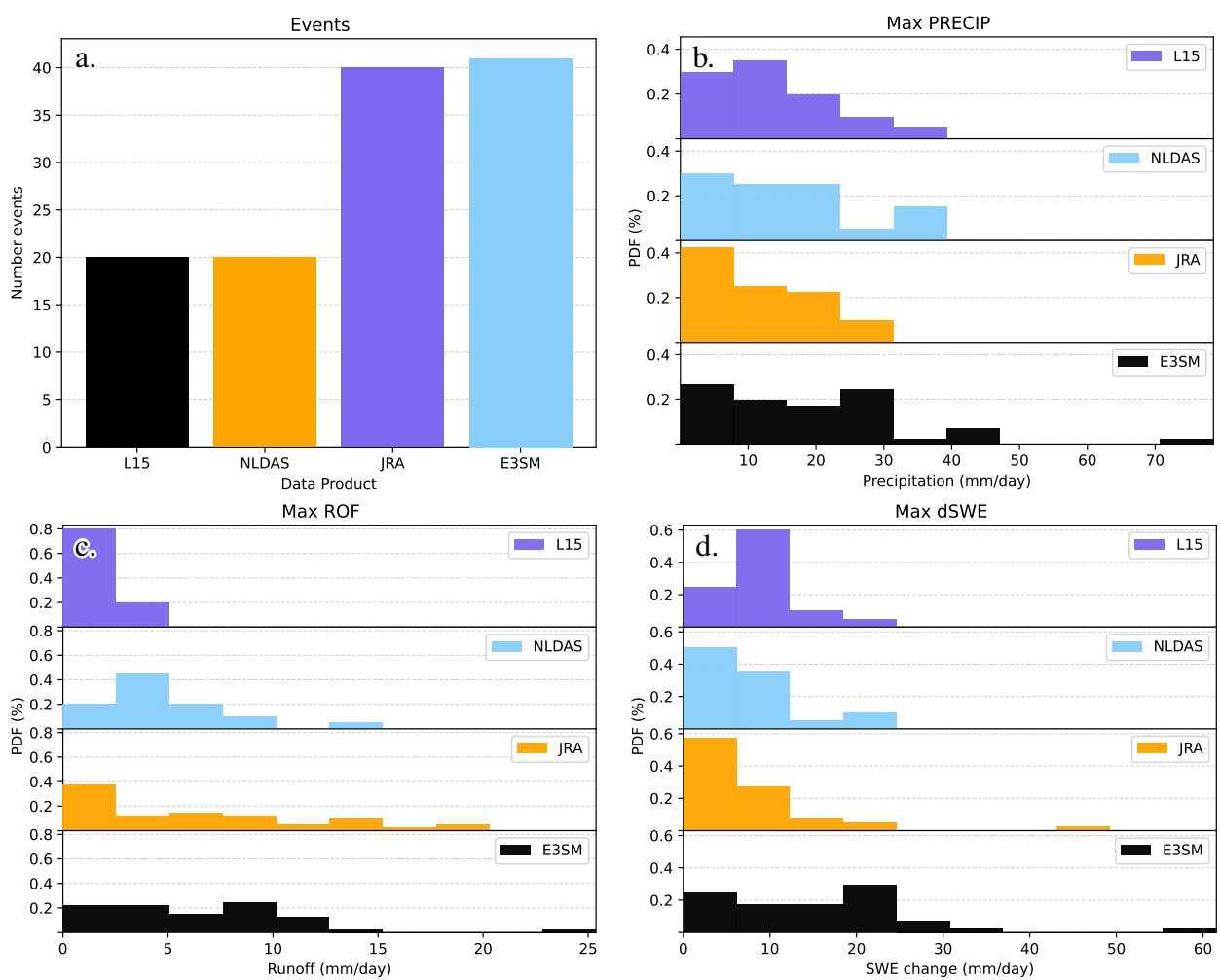

**Figure 4.** Same as Fig. 3 except for the RELATIVE detection thresholds.

The January 17-18, 1996 event is readily apparent for three of the four data products (NLDAS, JRA, and E3SM). Observed streamflow spiked during and immediately after the RoS event as shown by the dark blue stripes at the top of each time series, indicating a lag between when the RoS event algorithm identifies changes in PRECIP, dSWE, and ROF and the observed daily streamflow gage observation at Harrisburg (namely, the 99.9th percentile streamflow exceedance). While this is operationally the largest RoS event observed over the period, there remain large discrepancies in the hydrometeorological conditions between the datasets. The dSWE signal is the largest in magnitude within JRA and E3SM compared to NLDAS. L15 shows a very minimal dSWE, to the point that $t_{\text{dSWE}}$ is not exceeded in the detection algorithm. PRECIP is more similar across the three datasets, implying that the reduced ROF in L15 is likely due to minimal snowmelt.

All products also indicate a second, more moderate RoS event during the last third of February, although L15 and E3SM (JRA) show larger dSWE (ROF) signals than the other datasets. The USGS gage streamflow also exceeded the 95th percentile

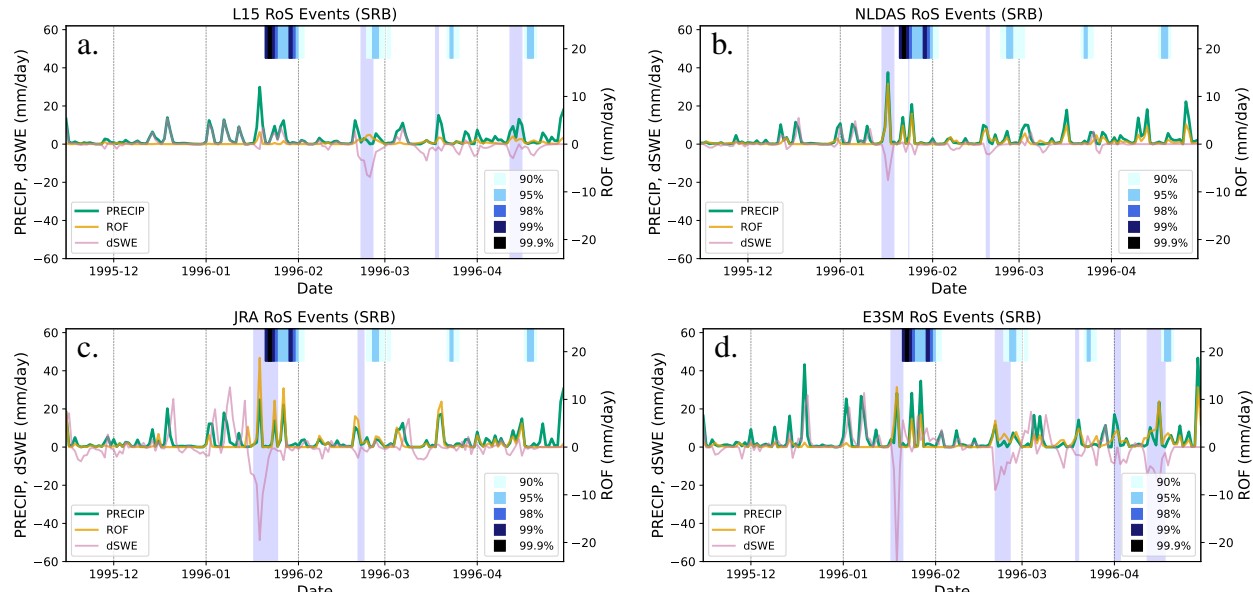

**Figure 5.** Time series of daily precipitation (PRECIP; green), surface runoff (ROF; orange), and SWE loss (dSWE; pink) during WY96 for each of the four products. Vertical purple shading denotes periods flagged as an RoS event. At the top of each time series are colored bars that denote extreme values (greater than the 90% percentile) of observed USGS streamflow at Harrisburg, Pennsylvania (#01570500), with black representing 99.9th percentile values over the 1980-2005 period.

for this RoS event. More moderate flooding also appeared to occur in March and April although the detection algorithm only flagged such events in L15 and E3SM. We also emphasize that all RoS events flagged by the detection algorithm resulted in well-above-average streamflow, highlighting the efficacy of the detection algorithm in capturing meaningful hydrometeorological extremes from basin-scale climate data.

Another way to visualize the WY96 RoS events is shown in Fig. 6. This plot shows the daily distribution of all three hydrometeorological quantities that make up a RoS event by plotting dSWE on the x-axis, PRECIP on the y-axis, and the size of the x-y marker depicts the magnitude of ROF. Markers are filled if the RoS event criteria are satisfied (RELATIVE) for that calendar day and are colored red (blue) if there is dSWE loss (gain). The variability of the PRECIP (y-axis) is the most similar between the two ESMs, although all products include at least one day of precipitation greater than 30 mm when averaged

over the SRB. Larger differences across datasets are noted for the other two RoS event response variables. JRA and E3SM produce a wider spread in both RoS and non-RoS events along the x-axis, indicating that the two ESMs produce days with the largest magnitudes of dSWE. Further, L15 has markers that are small in size, indicating low daily ROF magnitudes when compared to the other three products. There is a substantial spread across the four datasets, even over comparable time series with well-defined and record-setting RoS events. This is likely due to daily differences in the timing of storms, the amount of

water vapor transport and precipitation they produce, and how close surface air temperatures are to freezing conditions that in turn influence if a meteorological event leads to an increase in snow accumulation or results in an RoS event.

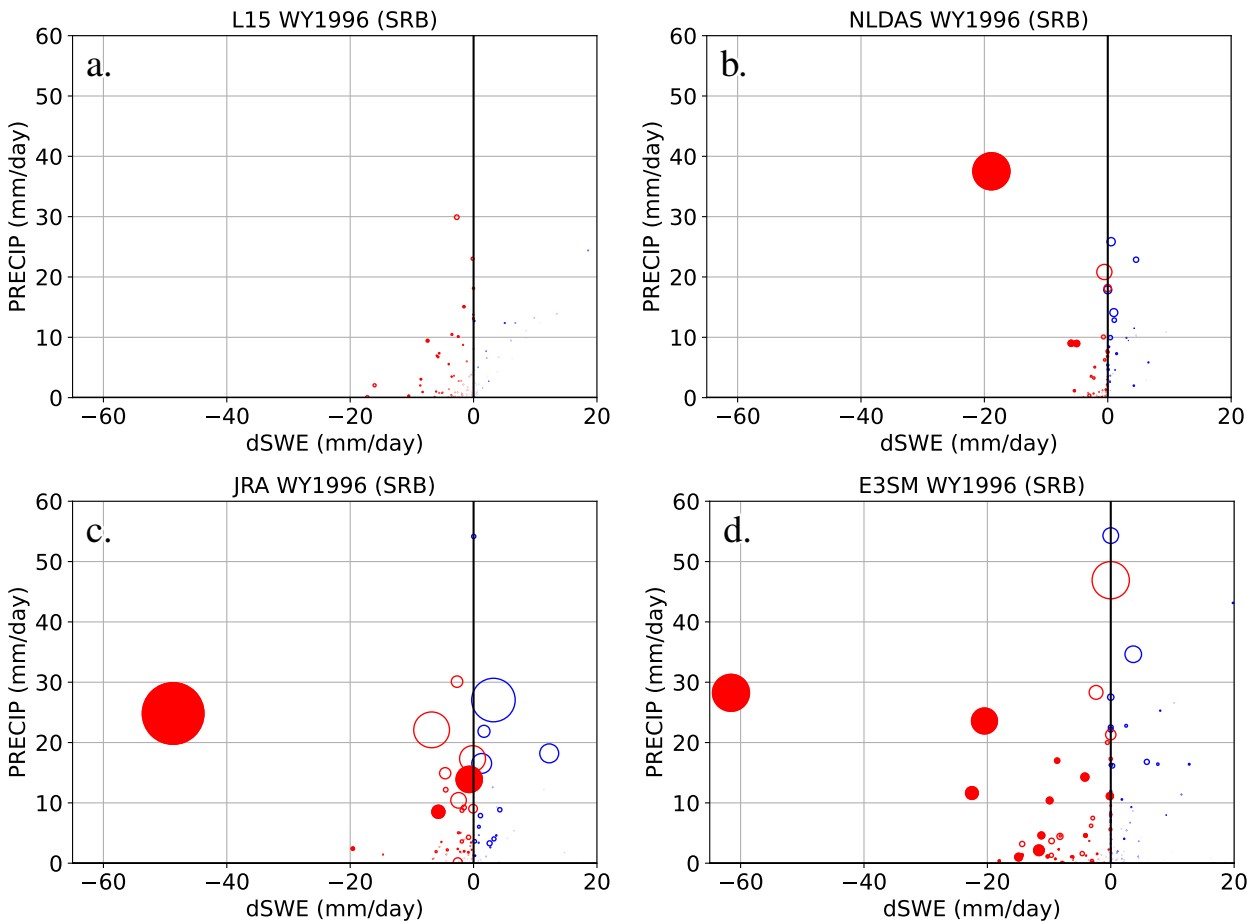

**Figure 6.** Bubble plots comparing precipitation (PRECIP; y-axis), SWE change (dSWE; x-axis), and surface runoff (size of bubble) for each of the four datasets during WY96. For each bubble, the values come from the same calendar day. Filled bubbles were flagged by the algorithm as an RoS event (purple shaded periods in Fig. 5).

## 3.4 Evaluation of a single event

One notable question given the above analysis (Fig. 5) is why the January 1996 event is captured differently across the products, particularly 'why is it 'unseen' by the L15 dataset?' Fig. 7 shows the SRB evolution of total on-surface SWE over WY96 for all datasets. All products show increasing SWE from mid-December onwards, albeit with varying accumulation rates and maximum depths. The datasets most differ in the SWE ablation during mid-January, with L15 (Fig. 7a) showing essentially no ablation of the existing snowpack, JRA (Fig. 7c) showing near total ablation, and NLDAS and E3SM (Fig. 7b,d) lying somewhere between the two extremes.

To untangle the potential source of this discrepancy, Fig. 8 shows the temperature (red line, right axis) at the grid cell nearest Harrisburg for the four products during the 1996 event. Also included is the temperature trace observed at Harrisburg (thin

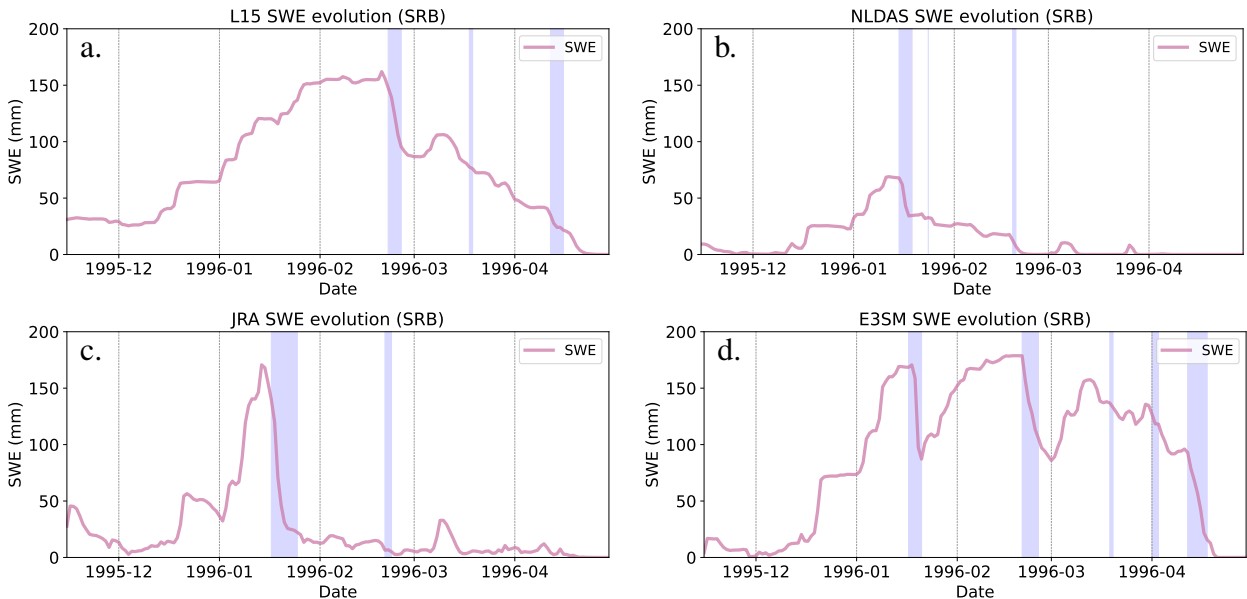

**Figure 7.** Daily SWE timeseries averaged over the SRB during WY96 for all four data products. Shaded in blue are the same events from Fig. 5.

gold line) as recorded in National Oceanic and Atmospheric (NOAA) surface weather station archives along with a reference freezing line (black). The surface relative humidity is indicated by the dashed purple line (right axis) and the product-specific associated precipitation rate (mm/hr) partitioned into rain (green) and snow (cyan) is denoted by bars (left axis).

None of the four products precisely match the observed temperature, although we acknowledge that this is, at least, partly due to the spatial resolution of the data. While the timing of the temperature increase associated with the warm air advection component of the event is well matched in NLDAS (Fig. 7b) and E3SM (Fig. 7d), both products have a slight cool bias. The overall temperature maximum is better matched by JRA (Fig. 7c), although the peak occurs approximately 6 hours earlier than in observations. An opposite shift is seen in L15 data, with the observed temperature peak leading L15's temperature peak by approximately 6 hours (Fig. 7a).

Across all four datasets, only L15 reports below-freezing temperatures between January 18th at 12Z and January 19th at 18Z. This occurs because L15 only leverages daily minimum and maximum observed temperatures to construct temperature fields. At each L15 grid cell, temperature minima and maxima are assumed to occur at sunrise and approximately late afternoon, respectively, with an interpolation spline used to provide information at intermediate hours (Bohn et al., 2013). These temperatures are always assumed to occur on the day of record (Livneh et al., 2015), which is relevant because the actual

temperature minimum on January 19th occurred during the local afternoon after the passage of a cold front (Leathers et al., 1998). A similar, regular cycle is seen in the L15 relative humidity, with near-saturated surface conditions only occurring for approximately 3-hour windows each day. The other three data products show the increased moisture advection ahead of the precipitation maximum, with near-saturated surface air mass for large periods of the 72 hours preceding the initiation of the

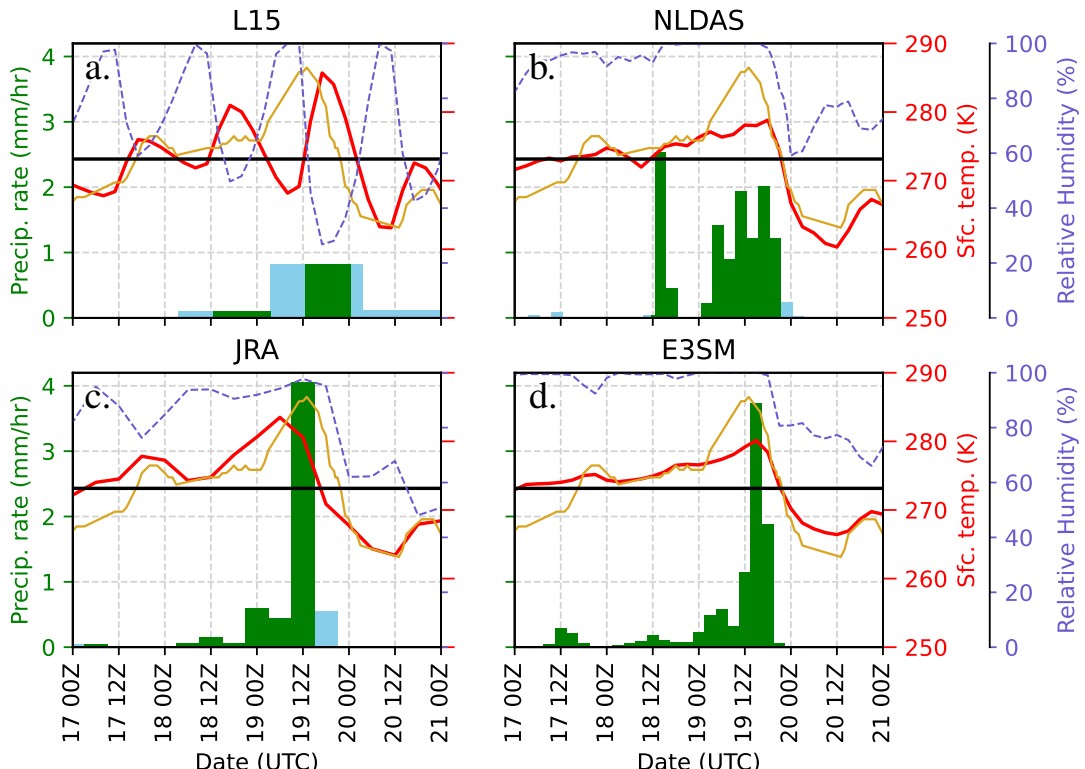

**Figure 8.** Evolution of reference height surface air temperature, relative humidity, and precipitation during the January 1996 SRB RoS flood event across data products. Data is taken from the grid cell nearest to Harrisburg, Pennsylvania. The left axis corresponds to the precipitation rate (shown in bars from the bottom) reported by the product. Green bars indicate rainfall (precipitation occurring when $T_s > 0°$C) and cyan bars indicate snowfall (precipitation occurring when $T_s <= 0°$C). The right axes show reference height surface air temperature as denoted by the solid red line, and relative humidity as denoted by the purple dashed line. A thick black horizontal line indicates $0°$C air temperature (273.15 K). The thin gold line on each panel represents the observed temperature trace at Harrisburg over the period.

flood event. Therefore, for this event, the diurnal cycle of thermodynamic variables is more strongly influenced by atmospheric
dynamics than solar insolation. We emphasize that this timing is actually quite important since factors such as temperature and surface humidity (and associated sensible and latent heat fluxes) are typically dominant drivers of snow ablation in high snowmelt events (Mazurkiewicz et al., 2008; Würzer et al., 2016; Harpold and Brooks, 2018) and need to line up concurrently to produce compound extremes like the one observed in 1996.

To understand how this could impact snowpack metamorphism, we also investigate precipitation during the event. We can
assume precipitation type at the surface is determined by whether the reference height surface air temperature is above or below freezing, which is the manner in which the majority of land surface models partition precipitation (Harpold et al., 2017; Jennings et al., 2018; Siirila-Woodburn et al., 2021a). Since L15 contains below-freezing temperatures between 00Z and 12Z on January 19, it is the only product that partitions a fraction of precipitation into the frozen state during the peak of the event.

Further, since daily precipitation is evenly partitioned into sub-daily bins to force VIC, L15 registers more precipitation during
this period than the other products (which typically peak after 12Z on January 19).

We argue this motivates the following interpretation. In L15, snowpack lost due to melt during the above-freezing temperatures during the 1996 event is, at least, partially offset by new snow on the ground that falls during the event due to below-freezing temperatures. These below-freezing periods and reduced relative humidity at the surface also mitigate any snowmelt in the land surface model that would be associated with enthalpy fluxes (either sensible or latent heat) into the ex-
isting snowpack. When combined, this explains the lack of a decline in L15 SWE and subsequent lack of a spike in ROF in Fig. 5. All other datasets show a more prolonged period of above-freezing temperatures with high specific humidity during the window noted above, with nearly all precipitation during this period falling as rain versus snow. This induced varying degrees of snowmelt in E3SM, NLDAS, and JRA that are sufficient to trigger the RoS detection algorithm.

## 3.5   Generalizability to other basins

Finally, while we focus on the SRB in this manuscript, it is beneficial to evaluate whether the algorithm can detect other well-known historical RoS events in the US and whether the dataset-to-dataset variability observed in the SRB occurs elsewhere. As a test of the transferability of the methodology described in Section 2.2, we perform the same analysis using the RELATIVE framework over the Willamette River Basin (WRB) in Oregon and southern Washington and Sacramento River Basin (SacRB), covering parts of Northern California and Southern Oregon in the United States. A significant RoS flood event occurred over
the WRB in 1996 a few weeks following the SRB event discussed above. From February 5th-9th, 1996, the WRB experienced its most severe flooding in three decades, with parts of the river rising up to 3-6 meters above flood stage, causing eight fatalities, displacing over 30,000 residents, and causing nearly $500 USD million in damages. Preceding the floods, subfreezing temperatures and substantial snowpack prevailed at relatively low elevations, but a succession of warmer synoptic systems brought liquid precipitation ranging from 10-25 cm in lowlands to 35-75 cm in mountainous regions over the four-day period.
These conditions led to significant snowmelt, exacerbating the flooding (Halpert and Bell, 1997; Colle and Mass, 2000). The 1997 New Year's flood event in California was the most financially devastating flood in the state's history with damages totaling $1.6 USD billion, ranked as the second most severe superflood between 1950 and 2010 across the western United States (Tarouilly et al., 2021). Over half a million individuals were displaced, and 43 out of 58 California counties were declared disaster zones (Lott et al., 1997). Preceding storms in late November and December contributed to elevated soil
moisture and a substantial snowpack, setting the stage for extreme flooding exacerbated by heavy precipitation and an intense, warm, atmospheric river event on New Year's Day of 1997 (Galewsky and Sobel, 2005; Rhoades et al., 2023).

To probe these events, we first acquire shapefiles defining both the WRB and SacRB, shown in Fig. 9. Like with the SRB analysis, all four datasets are masked using these basins. Daily, basin-averaged mean timeseries are constructed with the 95% percentiles of ROF and dSWE being computed from these distributions to use as thresholds $t_{ROF}$ and $t_{dSWE}$. We then seek RoS
events by looking for periods of concurrent PRECIP, ROF, and dSWE that all exceed relevant thresholds.

Fig. 10 shows the WY96 results over the WRB. The February flood event is clearly evident in the streamflow shading along the top of the panels on the left (black colors indicating 99.9% streamflow). Three of the four products indicate a RoS event

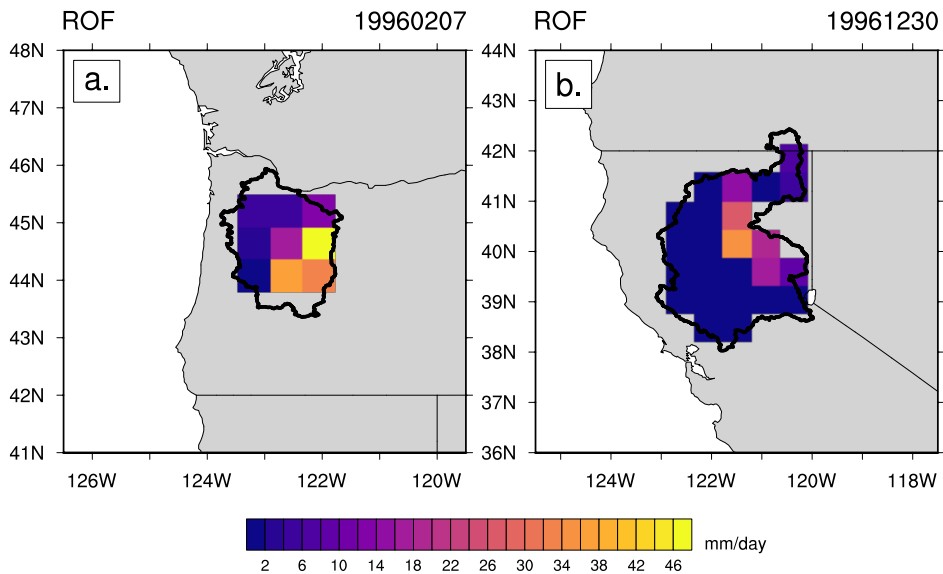

**Figure 9.** Basin shapefile domains for (a.) the Willamette River Basin (WRB) and (b.) the Sacramento River Basin (SacRB). Contoured is the surface runoff field from JRA on (a.) February 7th, 1996, and (b.) December 30th, 1996.

immediately preceding this streamflow maximum. All products indicate spikes in PRECIP and ROF over the basin, although E3SM produces less precipitation than the other three, likely owing to its coarser resolution. JRA has the largest and most rapid snowmelt, with both NLDAS and E3SM also showing reductions in SWE during the event. L15 shows a small rapid increase, then a decrease in SWE during the event. This offset is strong enough that the RoS algorithm is not triggered ($t_{dSWE}$ is not satisfied). While a more detailed evaluation of the meteorology as represented by the data products is beyond the scope of this study, it is possible that some of the mechanisms discussed in Section 3.4 are also relevant in this basin. The bubble plots on the right side of Fig. 10 show a wide diversity as in Fig. 6, although the relative differences are somewhat dissimilar, implying different processes are at play, particularly for the ROF. As before, L15 has a narrower range of dSWE over the basin (i.e., spread on the horizontal axis) but does contain large values of both precipitation (vertical axis) and runoff (marker size). The dSWE variability for both E3SM and JRA is larger but the products generally have lower ROF and PRECIP, respectively, than both L15 and NLDAS.

Figure 11 shows the WY97 results over the SacRB. Here, all four data products detect a RoS event in the basin in late December/early January. The timing differs slightly between the datasets, with E3SM (L15) triggering the earliest (latest). We speculate that this is a function of resolution, where the higher-resolution L15 contains more detailed small-scale processes (e.g., sub-basin melt at higher elevations, time for headwaters to reach the main stem). Evaluating temporal differences at the daily scale due to model structural characteristics is an interesting target for future work. All products capture large spikes in PRECIP and subsequent increases in ROF over the basin. They differ more significantly in terms of dSWE, with JRA (NLDAS) producing the most (least) snowmelt over the basin. Of note, both JRA and E3SM flag a smaller RoS event later in January

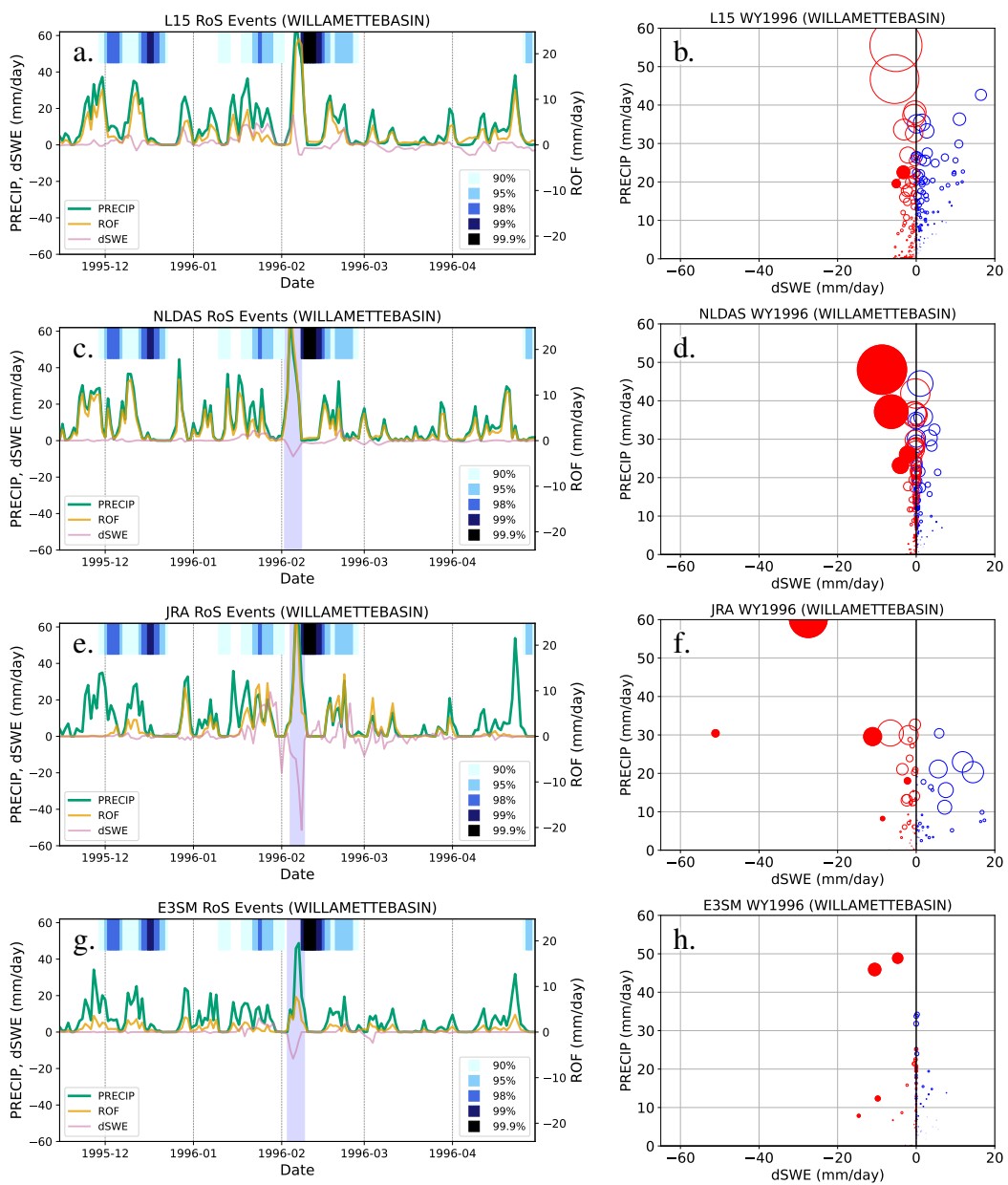

**Figure 10.** As in Figs. 5 (a., c., e., g.) and 6 (b., d., f., h.), except for the Willamette River Basin (WRB) during WY96. The striped gauge percentiles along the top of the left panels are derived from daily streamflow from USGS gage #14211720 (Willamette River at Portland, Oregon).

while both L15 and NLDAS show increased SWE in the basin. All products show increases in ROF, however, which coincide with a secondary maximum in the streamflow, highlighting the complexity of how surface hydrology evolves in models even if the relevant metrics (e.g., runoff) are similar. The bubble plots on the right of Fig. 11 indicate similar behavior to Fig. 10, which is likely a function of snow processes in the SacRB being more similar to the WRB than the SRB.

While this should only be considered a cursory investigation, it provides additional data points that the methodology can be applicable to other basins susceptible to RoS flooding. Of note, both the WRB and SacRB contain higher and steeper orography than the SRB, implying that the algorithm can credibly detect RoS events in different basin geometries. However, we stress a few caveats. More rigorous evaluation would be required by regional hydrologists to ensure results in other basins are hydrologically fit-for-purpose. Further, we also acknowledge that RoS over high mountain ridges can feed into rises in streamflow in different basins (e.g., the Sierra Nevada mountains in California contribute to multiple watersheds). Applying shapefiles containing larger hydrologic units or other spatial coverage may improve results in these regions and provide a better understanding of model variability and how land surface processes are represented in climate data.

## 4    Conclusions

We interrogate four different gridded climate data products that include PRECIP, ROF, and SWE in order to quantify differences in the representation of coupled land-atmosphere processes that lead to flooding events in the historical record. In particular, we focus on RoS floods over the SRB and devise an algorithm for the automatic detection of RoS events that can be applied to any gridded dataset with relevant variables. We include surface runoff as a criterion to incorporate information regarding the land surface model soil conditions before and during RoS events. We also spatially integrate at the basin scale versus the more commonly applied strategy of gridcell-by-gridcell analysis in climate data.

A detection algorithm flagging for times of collocated ROF and dSWE is generally successful at marking periods that will be followed by above-normal streamflow as measured by a gage in the basin. We find using fixed thresholds for RoS-relevant variables applied uniformly across multiple gridded datasets leads to large discrepancies in event frequency over the historical period, up to a factor of 10. Normalizing these thresholds by each dataset's climatology (relative thresholds based on daily percentiles) improves agreement. However, there remains approximately a factor of two difference between event counts, implying that the underlying distributions are fundamentally different in both shape and magnitude across the data analyzed.

This underscores the complex assessment of such flood events – even products representing the historical record contain a large spread in hydrometeorological quantities provided to end users. The spatiotemporal dependencies of how meteorological data is generated for land surface and/or hydrological model forcing are critically important. While spatial resolution of data is important, particularly for snow processes in complex terrain (Henn et al., 2018; Siirila-Woodburn et al., 2021a), we show that the time resolution of data used to derive surface water conditions is just as critical for transient extremes, such as RoS events. This is because the timing of synoptic variations in reference height surface air temperature and humidity that dictate snowmelt and precipitation phase can occur on the order of hours at local scales and can have an outsized role in compound

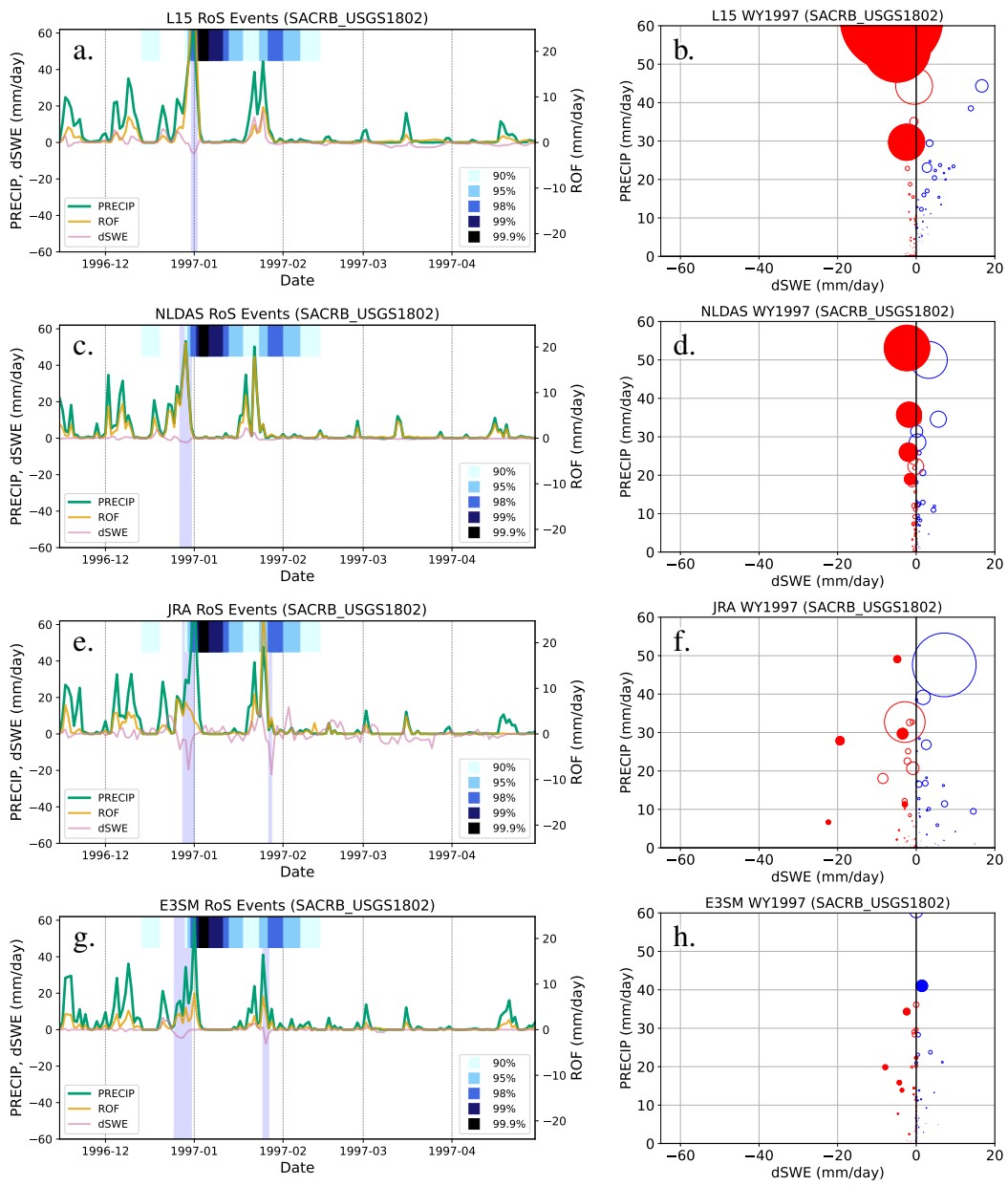

**Figure 11.** As in Figs. 5 (a., c., e., g.) and 6 (b., d., f., h.), except for the Sacramento River Basin (SacRB) during WY97. The striped gauge percentiles along the top of the left panels are derived from daily streamflow from USGS gage #14211720 (Sacramento River at Verona, California).

extreme event representation if those variations are threshold-dependent (e.g., storm rain-snow partitioning and alterations to the freezing line at the land surface).

Since RoS events in regions of ephemeral snow (e.g., SRB) can have rapid changes in surface forcing at hourly timescales (e.g., Leathers et al. (1998)), temporally coarse data applied to force LSMs may result in an underprediction of RoS frequency. This may occur if time-interpolated data (and/or less frequent coupling) smooths and/or clips out short-term extrema and results in mismatched forcing or reduced day-to-day variability of multiple, co-dependent anomalies (e.g., temperature and precipitation extremes) required to accurately simulate historical, decision-relevant hydrometeorological extremes (e.g., 1996

SRB RoS event). We note that this effect can be independent of the model timestep itself. For example, the L15 data disaggregates precipitation data across sub-daily timesteps and uses a diurnal spline to reconstruct near-surface atmospheric conditions. While the LSM used is integrated at smaller timescales, the effective time resolution of the forcing data remains daily.

     We also show that even though a dataset is provided at much coarser spatial resolutions than is desired (e.g., JRA and E3SM), model-derived datasets that are more frequently coupled and/or constrained at shorter timescales (e.g., reanalysis

products and nudged ESMs) may produce more accurate land-atmosphere interactions and better representation of decision-relevant hydrometeorological extremes, particularly at basin (and larger) spatial scales. However, these products will likely suffer in the spatial representation of hydrometeorology in regions of high heterogeneity. The spatial resolution is also tightly linked to land surface flood processes within the basin. Higher resolution may better resolve headwater catchments and smaller geometries whereas coarser datasets may only be skillful at larger scales, even with more accurate forcing. Spatial scales are

also connected to hydrologic timescales. Smaller areas respond to increased liquid input more quickly than larger, downstream sections of a basin. Understanding how coupling frequency affects hydrologic climate data, alongside spatial and temporal characteristics, is a complex challenge, but may offer insights into improving dataset credibility.

     We want to emphasize that this work doesn't aim to suggest a 'superior' dataset writ large. For example, even though L15 does not capture the 1996 SRB flood as well as other products here, it has been an invaluable tool for studying climate ex-

415 tremes such as heat waves (Mazdiyasni and AghaKouchak, 2015), droughts (Pendergrass et al., 2020; Williams et al., 2020), and wildfires (Williams et al., 2019) (amongst others) in the US. Rather, we offer a few suggestions about leveraging hydrometeorological data for application purposes. It is recommended that gridded climate data developers consider the temporal resolution of land surface forcing if the representation of daily (or sub-daily) hydrometeorological extrema is desirable, particularly from a use-inspired or decision-relevant context (Jagannathan et al., 2021). While we do not downscale any datasets

in this study, it is likely that using different meteorological data to force the same land surface and/or hydrological model will result in vastly different predicted streamflows, particularly for RoS events when variables are spatiotemporally co-dependent and would be sensitive to any post-processing adjustments (e.g., mean climatological correction). When possible, sub-daily information about atmospheric conditions should be included in meteorological forcing data.

     From a stakeholder perspective, this is an important consideration when back-testing models, and, in particular, applying

such models to evaluate tail risks (e.g., 1-in-100-year flood events and how they may change in a future climate). The results here show longer tails in the fully-coupled ESM-derived climate data, which may impact return rates of extreme events, even if calibrated or bias-corrected after the fact. Therefore, care must be taken when applying data requiring coupling between the

atmosphere and land surface (and riverine) components, whether generated dynamically or statistically. While post-processing adjustments to the mean climatology may be desirable, these adjustments can alter decision-relevant hydrometeorological

extremes that reside in the tail of the distribution.

*Code and data availability.* L15 data was downloaded from the NOAA Administration Physical Science Laboratory, available at https://psl.noaa.gov/data/gridded/data.livneh.html. NLDAS-VIC4.0.5 data was downloaded from the NOAA National Centers for Environmental Prediction, available at ftp://ldas.ncep.noaa.gov/nldas2/retrospective/vic4.0.5. JRA-55 data was downloaded from the Research Data Archive at the National Center for Atmospheric Research, Computational and Information Systems Laboratory, available at https://doi.org/10.5065/

D6HH6H41. The version of E3SM run here was v2.0.0-alpha.2-1816-gf9cbe57a2 and is available at https://github.com/E3SM-Project/E3SM. ERA5 data used to nudge the E3SM solution was downloaded from the Copernicus Climate Data Store (CDS), available at https://www.ecmwf.int/en/forecasts/datasets/reanalysis-datasets/era5. Station data for Harrisburg, PA was acquired from the Iowa Environmental Mesonet at https://mesonet.agron.iastate.edu/. The SRB is defined using a shapefile provided by the SRB Commission (https://www.srbc.net/portals/susquehanna-atlas/data-and-maps/subbasins/index.html). All processed data, figures, and scripts used to generate the results contained in this

manuscript (along with README documentation) are archived at Zenodo and can be accessed at https://doi.org/10.5281/zenodo.10412332.

*Author contributions.* CZ devised the project and led the writing of the manuscript, with input from RM and AR. BA post-processed climate data and wrote the majority of the RoS detection algorithm. AR performed an initial cursory analysis and devised the bubble plot visualization.

*Competing interests.* The authors declare no competing interests.

*Acknowledgements.* This work is supported by the U.S. Department of Energy (DOE), Office of Science, Office of Biological and Environ-

mental Research program under Award DE-SC0016605 "A framework for improving analysis and modeling of Earth system and intersectoral dynamics at regional scales." Data acquisition and E3SM simulations were completed at the National Energy Research Scientific Computing Center (NERSC), a U.S. Department of Energy Office of Science User Facility located at Lawrence Berkeley National Laboratory, operated under Contract No. DE-AC02-05CH11231 using NERSC award BER-ERCAP0020801. Event algorithm development, calibration, and analysis were performed on the Pennsylvania State University's Institute for Computational and Data Sciences' Roar supercomputer. The authors

thank Dr. Andrew Jones for initial thoughts regarding the bubble plot diagnostics contained in this paper. The authors also thank Dr. Keith Musselman and an anonymous reviewer for suggestions during the peer review process that improved this work.

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
