# Peer review of "Algorithmically Detected Rain-on-Snow Flood Events in Different Climate Datasets: A Case Study of the Susquehanna River Basin"

_EGUsphere, 2023_

## Author Comment (AC1)

**Author Response to Reviewer #2 of egusphere-2023-3094, 'Algorithmically Detected Rain-on-Snow Flood Events in Different Climate Datasets: A Case Study of the Susquehanna River Basin'**

Colin M. Zarzycki, Benjamin D. Ascher, Alan M. Rhoades, and Rachel R. McCrary

June 20, 2024

Specific responses regarding Reviewer #2's comments are contained in the text that follows. We have highlighted in areas in orange (chosen for color blindness) where we propose to further refine the manuscript in response to these comments.

**Response to Reviewer #2**
* * *
*The authors present an inter-model comparison of basin-scale rain-on-snow (RoS) flood events identified using different detection algorithms for a 21-year historical period over the Susquehanna River Basin, located in the eastern US. Compared to observations, the algorithm flags known historical events, but there are model-specific discrepancies in terms of event magnitude and the driving factors of precipitation, runoff, snowpack, and snowmelt contribution. The authors report that the use of dataset-specific thresholds improves agreement between models but does not account for all discrepancies, which are attributed to differences in model structure, meteorological forcing, and coupling frequency.*

*The paper is well-written and clearly structured. The references cited are complete and up to date. The results are relevant to local and regional flood risk assessments using gridded climate and hydrometeorological data sets that are available at continental to global scales.*

*I have two general comments and a few detailed edits.*

**Reply**: Thank you for your time and consideration in this review, please see our responses below, which we hope address all of this point-by-point and reflect an improved manuscript.
* * *
*It is mentioned that the ROF differences among hydrologic models can be extremely variable, with regional differences between products reaching an order of magnitude. Is this due to soil parameterization in models that influences infiltration capacity? Is there a difference in soil parameterization that sets L15 apart from the other models? Given the high uncertainty in soil parameterization, I wonder if there is benefit in including ROF as a threshold.*

*There is a misunderstanding evident on Line 125. The statement "20% reduction in the snowpack (implied dSWE <= -20 mm/day)" is incorrect. The 20% snowmelt contribution threshold applied by Freudiger et al. (2014), which was subsequently adopted by Musselman et al. (2018) and Li et al. (2019), is described as: "... for which the sum of rainfall and snowmelt contains a minimum of 20% snowmelt." This is a key point to make clear. What is helpful about this threshold condition is that it doesn't rely on characterization of soil infiltration capacity, which is difficult to validate in models and can vary over an order of magnitude regionally and between models.*

*What I like about your inclusion of the ROF variable is that it connects to the importance of antecedent soil moisture and saturated conditions to driving flood response. More discussion on this topic is probably warranted. Requiring ROF>0 is a condition that the soil must be saturated to result in flood, which is intuitive, and then not surprising that the flagged events in the simulations correspond to streamflow events in the USGS record.*

*Given that (at least) these three papers adopted the "20%" threshold to identify times when snowmelt is substantially contributing to rainfall + snowmelt fluxes, I encourage the authors to include this 20% value in their FIXED (and X% in their RELATIVE) analyses. Given the uncertainty associated with estimating ROF, can you identify flood events in the record without requiring ROF? What is potentially missed / lost if one doesn't include ROF? For example, excluding ROF as a condition is analogous to not knowing the buffering capacity of soil water deficit. Would that impact the ability of an automated algorithm to flag potentially impactful RoS events? This would seem to be an important tie to previous studies – because I do think antecedent soil saturation is important - and could result in key guidance to future research efforts.*

**Reply**:  The mischaracterization of Freudiger et al. [2014] is a somewhat embarrassing oversight on our part. We have updated the text to reflect the criterion that at least 20% of the sum of snowmelt and rainfall must come from snowfall (we note that the ratio of rainfall to snowmelt cannot exceed 4:1, which is mathematically identical).

We appreciate the suggestion. Because this is an interesting question, we have performed a sensitivity analysis where we also impose this threshold in our algorithm. We define a new threshold $t_{\mathrm{fSWE}}$ equal to 0.2, which is computed as dSWE (sign flipped for snowmelt to be positive) divided by the sum of dSWE and PRECIP. As with the other three thresholds, $t_{\mathrm{fSWE}}$ must be satisfied in our sensitivity run (RELATIVE_F14). We find that for three of the four datasets, there is no change in the number of events. The one dataset that sees reduced event frequency is NLDAS, a product that shows high climatological PRECIP and low climatological SWE (and dSWE) over the SRB. Even though event frequency is the same we see a slight reduction in event length and average ROF in the other three datasets, implying that some days at event onset or event termination are removed when $t_{\mathrm{fSWE}}$ is used.

We add these results to the text for readers.

We also perform a sensitivity analysis intended to include the requirement in Freudiger et al. [2014] and Musselman et al. [2018] that the sum of rainfall and snowmelt contains at least 20% snowmelt. This is added as an additional threshold $t_{\mathrm{fSWE}} = 0.2$ where fSWE (fraction of dSWE contribution) is computed as dSWE divided by the sum of dSWE and PRECIP smoothed time series (note, the sign of dSWE is inverted to be a positive contributor to liquid water on the surface). We refer to this simulation as RELATIVE_F14 since it preserves the same thresholds in RELATIVE with this one additional exclusionary check from Freudiger et al. [2014] in order to remove high rainfall (but low snow loss) events. The number of events is the same for all datasets except NLDAS, which loses 7 events over the study period when enforcing $t_{\mathrm{fSWE}} = 0.2$. This can be explained by the results in Fig. 2. NLDAS produces a 'wetter' precipitation climatology (Fig. 2f) but less climatological SWE (Fig. 2j) and, correspondingly, less dSWE (Fig. 2n). Therefore, enforcing a check that removes high PRECIP, low dSWE events would reduce events detected in NLDAS most strongly relative to other datasets. While the other three products have the same number of events with or without the inclusion of $t_{\mathrm{fSWE}} = 0.2$, the mean duration is slightly shortened, and mean event ROF is somewhat reduced using RELATIVE_F14, implying that a handful of high PRECIP, low dSWE (and high ROF) days at event onset or termination are lost. However, this reduction is small and therefore provides confidence that just including threshold checks for dSWE, ROF, and PRECIP is satisfactory for detecting RoS events over the SRB without a more formal snow loss cutoff. In the remainder of this paper, we omit the use of $t_{\mathrm{fSWE}}$ for simplicity. However, we want to emphasize that the simulation of land surface processes in different datasets can play a key role in precipitation/snowmelt partitioning, motivating further process-oriented evaluation of their joint occurrence in gridded climate data in the future. A percentile-based threshold-only algorithm (such as RELATIVE) may particularly struggle in regions of climatologically low SWE and high PRECIP (which experiences primarily rain-induced flooding) or high SWE and low PRECIP (primarily melt-induced flooding).

The updated table reads:

Table R1: Statistics of RoS events over the SRB using FIXED thresholds (top) and RELATIVE thresholds (bottom). $t_{\mathrm{PRECIP}}$, $t_{\mathrm{ROF}}$, and $t_{\mathrm{dSWE}}$ represent the thresholds used for precipitation, surface runoff, and snow water equivalent loss, respectively (mm/day). Events represent the number of RoS events flagged over the 1980-2005 period. Duration is the average number of consecutive days an RoS event lasts. $\mathrm{PRECIP}_a$, $\mathrm{ROF}_a$, and $\mathrm{dSWE}_a$ represent the amount of precipitation rate, amount of runoff rate, and average snow loss (mm/day) per event by calculating the mean daily value for each individual event and then averaging those.

| | $t_{\mathrm{PRECIP}}$ $mm\ day^{-1}$ | $t_{\mathrm{ROF}}$ $mm\ day^{-1}$ | $t_{\mathrm{dSWE}}$ $mm\ day^{-1}$ | $t_{\mathrm{fSWE}}$ % | Events # | Duration days | $\mathrm{PRECIP}_a$ $mm\ day^{-1}$ | $\mathrm{ROF}_a$ $mm\ day^{-1}$ | $\mathrm{dSWE}_a$ $mm\ day^{-1}$ |
|---|---|---|---|---|---|---|---|---|---|
| FIXED | | | | | | | | | |
| L15 | 2.0 | 1.4 | -1.4 | - | 6 | 3.2 | 9.5 | 1.5 | -5.3 |
| NLDAS | 2.0 | 1.4 | -1.4 | - | 16 | 2.9 | 6.3 | 2.1 | -5.2 |
| JRA | 2.0 | 1.4 | -1.4 | - | 48 | 3.5 | 5.3 | 2.6 | -4.4 |
| E3SM | 2.0 | 1.4 | -1.4 | - | 58 | 4.1 | 6.5 | 2.4 | -5.2 |
| RELATIVE | | | | | | | | | |
| L15 | 2.0 | 0.6 | -1.5 | - | 20 | 5.2 | 6.2 | 1.1 | -5.3 |
| NLDAS | 2.0 | 1.4 | -0.8 | - | 20 | 2.8 | 7.2 | 2.1 | -4.2 |
| JRA | 2.0 | 1.6 | -1.9 | - | 40 | 3.1 | 5.1 | 2.5 | -4.1 |
| E3SM | 2.0 | 1.8 | -2.2 | - | 41 | 4.1 | 6.9 | 2.9 | -8.0 |
| RELATIVE_F14 | | | | | | | | | |
| L15 | 2.0 | 0.6 | -1.5 | 20 | 20 | 4.9 | 6.0 | 1.0 | -5.3 |
| NLDAS | 2.0 | 1.4 | -0.8 | 20 | 13 | 3.0 | 4.4 | 1.9 | -5.1 |
| JRA | 2.0 | 1.6 | -1.9 | 20 | 40 | 3.0 | 5.3 | 2.4 | -4.2 |
| E3SM | 2.0 | 1.8 | -2.2 | 20 | 41 | 4.0 | 6.8 | 2.8 | -7.8 |

In general, we find that the algorithm is largely insensitive to this inclusion, although we add additional text suggesting further evaluation may be merited in different cases (for example, datasets and/or regions with a high frequency of ROF events arising from high PRECIP). In some ways, this – to us – implies using runoff and snowmelt is therefore implicitly enforcing events to contain high snowmelt fraction in a process sense. However, we choose not to speculate regarding that in the manuscript without further (future) investigation beyond the scope of this project.
* * *
*It would be helpful to see more written about the transferability of the results to other regions / basins. Particularly, how transferrable would your results be to regions with more elevation complexity as it relates to model treatment of elevation-dependent hydrometeorology, snowpack, and soil antecedent conditions, which would seem relevant to the differences in model grid spacing? If not transferrable, could the authors provide guidance for future research efforts?*

**Reply**: This was also suggested by Reviewer #1, so we have included a new section in the manuscript entitled 'Generalizability to other basins.' We have reproduced this new text in the Appendix of this response. In this section, we explore two basins in the US with notable RoS events (Willamette River Basin and Sacramento River Basin). Both basins contain more orographic complexity than the SRB. While only a cursory analysis, the algorithm did detect the largest RoS event over the study period in each basin. We find this result promising, although we also noted potential caveats and emphasized that this could be (an interesting!) target for future work, particularly for researchers interested in including ROF in RoS algorithms and/or exploring flood responses at the basin scale versus gridpoint values.
* * *
*Line 181: "(i.e., positive dSWE in both figures denotes snowpack)", I think "denotes snow accumulation" is more accurate.*

**Reply**: Typo corrected.
* * *
*Line 225: This should be "WY 1996". In the US, the water year is designated by the calendar year in which it ends. The year ending September 30, 1996 is called the "1996 WY", which begins on Oct 1, 1995.*

**Reply**: Another embarrassing oversight. These have been corrected.
* * *
*Figures 2 & 3: The histograms in panels b-d are very difficult for me to visually disentangle. Please consider making each variable-panel into 2x2 sub-panels showing the individual (solid / filled) colored histograms for each model.*

**Reply**: We agree that these plots were difficult to interpret. We liked this suggestion to create subpanels and it spurred an idea to convert these plots into 4x1 subpanels. We believe this does provide an improved interpretation of the previously overlapping plots. The updated version of Figure 3 is shown below.

[Figure]

Figure R1: (Fig. 3 from manuscript) Same as Fig. 2 except for the RELATIVE detection thresholds.

*Thank you for the chance to review this important work.*

**Reply**: Thank you for your thoughtful review and suggestions on further improving this case study.

**Appendix A:  Generalizability to other basins**

*Author note: The below block of code will be a new section in a revised manuscript.*

[revised manuscript text omitted]

N. Lott, M. C. Sittel, and D. Ross. The winter of '96-'97: West Coast flooding. Technical Report 97–01, National Climatic Data Center, 1997. URL https://repository.library.noaa.gov/view/noaa/13812.

K. N. Musselman, F. Lehner, K. Ikeda, M. P. Clark, A. F. Prein, C. Liu, M. Barlage, and R. Rasmussen. Projected increases and shifts in rain-on-snow flood risk over western North America. *Nature Climate Change*, 8(9):808–812, 2018. doi: 10.1038/s41558-018-0236-4.

A. M. Rhoades, C. M. Zarzycki, H. A. Inda-Diaz, M. Ombadi, U. Pasquier, A. Srivastava, B. J. Hatchett, E. Dennis, A. Heggli, R. McCrary, S. McGinnis, S. Rahimi-Esfarjani, E. Slinskey, P. A. Ullrich, M. Wehner, and A. D. Jones. Recreating the California New Year's flood event of 1997 in a regionally refined Earth system model. *Journal of Advances in Modeling Earth Systems*, 15(10):e2023MS003793, 2023. ISSN 1942-2466. doi: 10.1029/2023MS003793.

E. Tarouilly, D. Li, and D. P. Lettenmaier. Western U.S. superfloods in the recent instrumental record. *Water Resources Research*, 57(9):e2020WR029287, 2021. ISSN 0043-1397. doi: 10.1029/2020WR029287.

---

## Author Comment (AC2)

**Author Response to Reviewer #1 of egusphere-2023-3094, 'Algorithmically Detected Rain-on-Snow Flood Events in Different Climate Datasets: A Case Study of the Susquehanna River Basin'**

Colin M. Zarzycki, Benjamin D. Ascher, Alan M. Rhoades, and Rachel R. McCrary

June 20, 2024

Specific responses regarding Reviewer #1's comments are contained in the text that follows. We have highlighted in areas in orange (chosen for color blindness) where we propose to further refine the manuscript in response to these comments.

**Response to Reviewer #1**
* * *
*This study makes a useful contribution to studying rain-on-events (at ephemeral snowpacks) by developing and evaluating an algorithm that can detect basin-scale rain-on-snow events using gridded climate data. The method searches for periods of concurrent (area-averaged) precipitation, surface runoff, and snowmelt exceeding pre-defined values. Application of this to Susquehanna River Basin (SRB) indicates that (using dataset-specific thresholds) the method seems to work appropriately. The method also highlights there can be large differences in RoS event magnitude and frequency caused by differences in the various driving factors.*

*In principle, this paper could be published, as the developments seem useful, and are logically shown. However, at the same time, the sort of ad-hoc presentation of a single basin results makes it unclear how useful the methods are, and how it's usefulness varies between scales and settings. This would make the paper more useful and thereby a better fit for this journal.*

**Reply**: Thank you for agreeing to, and taking the time to, review this work. We have considered the suggestions below and believe addressing them has improved the manuscript.
* * *
*It would be useful if a clearer workflow is presented. This helps other people to better understand and potentially adopt the approach*

**Reply**: Thank you for the suggestion. We had added some text to better describe the methodology, but the biggest improvement is the addition of a visual schematic that highlights the steps required to take daily, gridded climate data and produce an RoS event climatology as we have done in this paper. We add the below text and figure to the manuscript to better describe the RoS algorithm.

A sample RoS event detection is shown in Fig. 1c. The smoothed basin-averaged daily time series of PRECIP, ROF, and dSWE are shown from top-to-bottom in dark green, blue, and red respectively (the thinner line represents the raw, unsmoothed time series). The various thresholds ($t_{\mathrm{PRECIP}}$, $t_{\mathrm{ROF}}$, and $t_{\mathrm{dSWE}}$) are shown as dashed horizontal lines. The area where the metric exceeds the relevant $t$ (i.e., days that the variable's RoS criterion is satisfied) is shaded for each time series. The vertical black lines denote the start and end of the event, defined by the first and last times when all three quantities exceed their defined threshold $t$. Gray shading represents contiguous days where all three criteria are satisfied, thus defining a RoS event. Here, an event from January 17th to January 25th, 1996 was added to the record.

[Figure]

Figure R1: Schematic demonstrating how RoS events are defined in this work. (a.) Gridded daily-average data is first masked to only retain data within a defined shapefile (here the SRB) and then area-averaged to produce a single value on that day for the area. (b.) Exceedance thresholds $t$ can be computed from these daily values by using a specified percentile (e.g., 95%) of the distribution of the entire dataset. (c.) Finally, RoS events are defined as contiguous days where the basin-averaged time series of ROF, dSWE, and PRECIP all exceed their thresholds $t_{\mathrm{ROF}}$, $t_{\mathrm{dSWE}}$, and $t_{\mathrm{PRECIP}}$, respectively. In the bottom plots, the darker lines represent the smoothed timeseries while the thinner lines denote the raw daily data. Shaded areas indicate periods when the given variable's time series is above its relevant threshold $t$ (denoted as a horizontal dashed line).
* * *
*The method is applied to one basin. This is OK, but also a bit a thin basis for introducing a method. Across what range of areas is your method likely applicable?*

*The comparison of climate data for a single basin (section 3.1) is useful, but it would be a lot more useful to systematically test how these things vary? (i.e. beyond this specific basin).*

**Reply**: This is similar to the suggestion from Reviewer #2 about generalizability. While a global (or even large-scale regional) evaluation of how well the technique transfers to other basins is beyond the scope of this work, we add a new section to the manuscript entitled '**Generalizability to other basins**.' We have reproduced this new text in the Appendix of this response.

In this section we test the RELATIVE configuration of the algorithm in two additional basins in the western United States: the Willamette River Basin (WRB) in Oregon and southern Washington and the

Sacramento River Basin (SacRB), covering parts of Northern California and Southern Oregon. We evaluate two water years where notable rain-on-snow events occurred – a flood in the WRB in February 1996 and one in the SacRB at the end of December 1996 into January 1997. We find that the algorithm detects these RoS events without additional modification/tuning, which we take as a positive sign that the framework could be extended to other basins. In addition to a cursory comparison of the four datasets, we also note that there are potential ways to further improve this framework by using larger hydrologic units, modifying the regions in which data is averaged over, etc.
* * *
*The timescale over which RoS are relevant will be very dependent on the spatial scale of the watershed. For example, headwater catchments may need timescales of hours whereas for large basins it may be multiple days. Can this be discussed?*

**Reply**: This is a good point. We have added additional text in the discussion section that notes that the findings here are tied to both the spatial and temporal scales of the hydrologic processes in addition to the forcing/coupling frequency.

"We also show that even though a dataset is provided at much coarser spatial resolutions than is desired (e.g., JRA and E3SM), model-derived datasets that are more frequently coupled and/or constrained at shorter timescales (e.g., reanalysis products and nudged ESMs) may produce more accurate land-atmosphere interactions and better representation of decision-relevant hydrometeorological extremes, particularly at basin (and larger) spatial scales. However, these products will likely suffer in the spatial representation of hydrometeorology in regions of high heterogeneity. The spatial resolution is also tightly linked to land surface flood processes within the basin. Higher resolution may better resolve headwater catchments and smaller geometries whereas coarser datasets may only be skillful at larger scales, even with more accurate forcing. Spatial scales are also connected to hydrologic timescales. Smaller areas respond to increased liquid input more quickly than larger, downstream sections of a basin. Understanding how coupling frequency affects hydrologic climate data, alongside spatial and temporal characteristics, is a complex challenge, but may offer insights into improving dataset credibility"
* * *
*Provide units at all axes go all figures.*

**Reply**: Thank you for catching that! We have carefully gone through the manuscript and all figures have been updated to better describe units on all axes.

**Appendix A: Generalizability to other basins**

*Author note: The below block of code will be a new section in a revised manuscript.*

[revised manuscript text omitted]

N. Lott, M. C. Sittel, and D. Ross. The winter of '96-'97: West Coast flooding. Technical Report 97–01, National Climatic Data Center, 1997. URL https://repository.library.noaa.gov/view/noaa/13812.

A. M. Rhoades, C. M. Zarzycki, H. A. Inda-Diaz, M. Ombadi, U. Pasquier, A. Srivastava, B. J. Hatchett, E. Dennis, A. Heggli, R. McCrary, S. McGinnis, S. Rahimi-Esfarjani, E. Slinskey, P. A. Ullrich, M. Wehner, and A. D. Jones. Recreating the California New Year's flood event of 1997 in a regionally refined Earth system model. *Journal of Advances in Modeling Earth Systems*, 15(10):e2023MS003793, 2023. ISSN 1942-2466. doi: 10.1029/2023MS003793.

E. Tarouilly, D. Li, and D. P. Lettenmaier. Western U.S. superfloods in the recent instrumental record. *Water Resources Research*, 57(9):e2020WR029287, 2021. ISSN 0043-1397. doi: 10.1029/2020WR029287.